

# The vegetation of Holocene coastal dunes of the Cape south coast, South Africa

Richard M. Cowling[1], Hayley Cawthra[1,2], Sean Privett[3] and B. Adriaan Grobler[1]

[1] African Centre for Coastal Palaeoscience, Nelson Mandela University, Gqeberha, Eastern Cape, South Africa
[2] Minerals and Energy Unit, Council for Geoscience (Western Cape Office), Cape Town, Western Cape, South Africa
[3] Grootbos Foundation, Grootbos Nature Reserve, Gansbaai, Western Cape, South Africa

## ABSTRACT

The vegetation of calcareous coastal dunes of Holocene age along the south coast of South Africa's Cape Floristic Region is poorly described. This vegetation comprises a mosaic of communities associated with two biomes, Fynbos and Subtropical Thicket. Previously, expert knowledge rather than quantitative floristic analysis has been used to identify and delimit vegetation units. In many areas, mapped units conflate vegetation on Holocene sand with that on unconsolidated sediments of late Pleistocene age, despite pronounced species turnover across this edaphic boundary. Despite dominance by Cape lineages and fynbos vegetation, dune vegetation in the eastern part of the region has been included in the Subtropical Thicket Biome rather than the Fynbos Biome. The high levels of local plant endemism associated with this dune vegetation and the small and fragmented configuration of these habitats, makes it an urgent conservation priority especially when placed in the context of rising sea levels, increasing development pressures and numerous other threats. Here we provide a quantitative analysis of 253 plots of the 620 km$^2$ of Holocene dune vegetation of the study area using phytosociological and multivariate methods. We identified six fynbos and two thicket communities based on the occurrences of 500 species. Following a long tradition in Cape vegetation typology, we used the Strandveld (beach vegetation) concept as our first-order vegetation entity and identified six units based on the fynbos floras. These were, from east to west, Southeastern Strandveld, St Francis Strandveld, Goukamma Strandveld, Southwestern Strandveld and Grootbos Strandveld. Each unit was differentiated by a suite of differential species, most being Holocene dune endemics. The two thicket communities—Mesic and Xeric Dune Thicket—showed limited variation across the study area and were subsumed into the Strandveld units. We discussed our findings in terms of vegetation–sediment relationships, emphasizing the need for a greater geographical coverage of sediment ages to facilitate a better understanding of deposition history on vegetation composition. We also discussed the role of soil moisture and fire regime on structuring the relative abundance of fynbos and thicket across the Holocene dune landscape. Finally, we address the conservation implications of our study, arguing that all remaining Holocene dune habitat should be afforded the highest conservation priority in regional land-use planning processes.

Corresponding authors
Richard M. Cowling,
rmcowling27@gmail.com
B. Adriaan Grobler,
adriaan.grobler85@gmail.com

---

## INTRODUCTION

The sedimentary deposits associated with Holocene coastal dunes are regionally and globally rare habitats (*Roberts et al., 2006*), characterized—in relation to their immediate hinterlands—by harsh environments (*Hesp & Martínez, 2007*) and relatively species-poor, albeit highly endemic, floras (*Grobler et al., 2020*; *Grobler & Cowling, 2021*; *Kruckeberg & Rabinowitz, 1985*; *Van der Maarel, 1993*). They provide a wide array of ecosystem services, notably coastal protection, erosion control, water retention and purification, recreation and tourism, as well as various cultural benefits (*Everard, Jones & Watts, 2010*; *Garcia Rodrigues et al., 2017*; *Martínez et al., 2007*). Coastal regions are the most densely populated areas on Earth (*Small & Nicholls, 2003*), with human pressure on coastal environments increasing drastically in recent decades (*Williams et al., 2022*). This surge in human activities—urbanization, agriculture, forestry, industry, transport, and tourism—has resulted in a decline in the extent and quality of coastal dune habitats (*Martínez, Psuty & Lubke, 2008*). In addition to coastal development, the predicted upsurge in extreme climatic events and sea-level rise due to climate change further exacerbate the challenges faced by coastal dunes, causing increased fragmentation and loss of habitats (*Feagin, Sherman & Grant, 2005*). The protection and restoration of remnant coastal dunes is thus an urgent priority (*Tinley, 1985*).

Our focus here is on the vegetation of unconsolidated, calcareous coastal dunes of Holocene age on the south coast of South Africa's Cape Floristic Region (CFR) (Fig. 1). We excluded the vegetation of modern mobile dunefields and the backbeach zone, the latter comprising dwarf shrublands and herblands of frontal dunes, coastal cliffs and rocky coasts. These are treated comprehensively elsewhere (*Taylor & Boucher, 1993*). We also exclude vegetation growing on Late Pleistocene sediments and rocks, comprising unconsolidated and cemented dunes (aeolianites) (*Roberts et al., 2006*; *Cawthra et al., 2020b*). The vegetation on these older surfaces differs markedly from that on the Holocene sediments (*e.g.*, *Thwaites & Cowling, 1988*).

The composition of fixed coastal dune vegetation of the Cape south coast and elsewhere is influenced by several geographical contingencies, namely dune area and topography (*Tinley, 1985*; *Cowling, 1984*), composition and ecology of hinterland vegetation (*Laliberté, Zemunik & Turner, 2014*; *Grobler & Cowling, 2021*), sediment age (*Thwaites & Cowling, 1988*; *Laliberté, Zemunik & Turner, 2014*) and the extent of available space for accommodating dune habitat on land exposed during glacial periods (*Grobler et al., 2020*). Long-distance oceanic dispersal plays a relatively minor role in structuring the vegetation of fixed Holocene dunes; most species are derived *via* colonization from hinterland floras (*Brunbjerg et al., 2014*; *Grobler & Cowling, 2021*).

The delimitation of vegetation units on coastal dunes on the Cape south coast is ripe for re-analysis. The current vegetation map (*Rebelo et al., 2006*; *Grobler et al., 2018*) is not based on formal floristic analysis and the mapped units frequently combine vegetation on Holocene dunes with that on older (Pleistocene and Neogene), more weathered and calcium-poor sands. Producing a defensible classification of the dune vegetation is important for habitat protection, since conservation priority status in South Africa is based

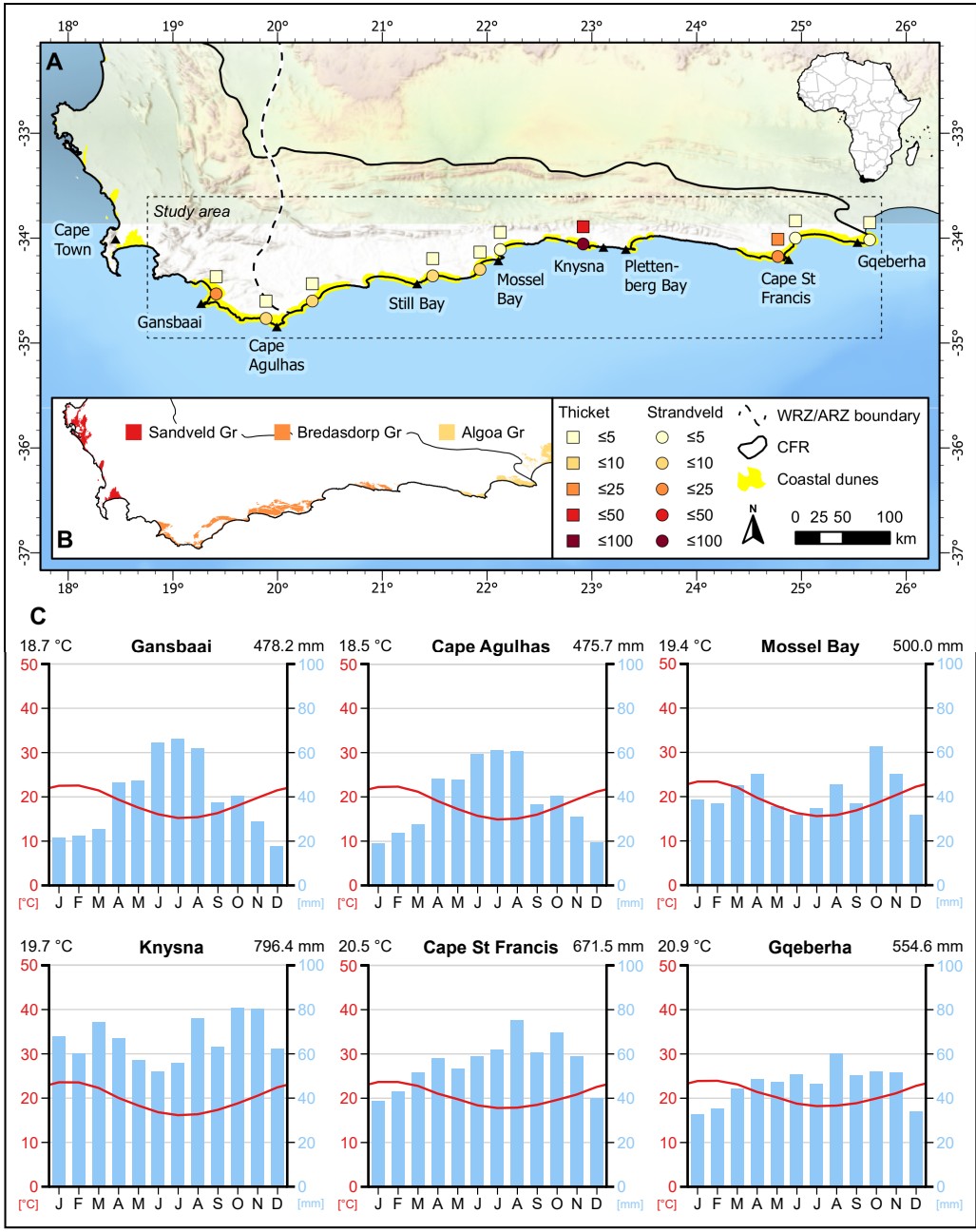

**Figure 1** **Distribution of sampling sites in coastal dunes of the south coast of the Cape Floristic Region (CFR).** (A) Number of sampling plots in thicket (squares) and fynbos (circles) communities of strandveld vegetation in the study area. (B) Distribution of Neogene to Recent coastal sedimentary deposits and corresponding geological groups in the CFR. (C) Climate diagrams representative of the sampling sites. Modified from diagrams produced at ClimateCharts.net (*Zepner et al., 2020*).

on the extent to which area-based targets can be or have been achieved in untransformed habitat (*Rouget et al., 2006*; *Botts et al., 2020*). Thus, vegetation types where there is no longer sufficient intact habitat to achieve a target (say 20% of the original extent of habitat)

emerge as conservation priorities, whereas those where untransformed habitat greatly exceeds the target, are regarded as "Least Threatened". It follows that an accurate map of vegetation types influences significantly their conservation status.

The extent of Cenozoic sediments in the CFR is inadequately delimited (*Botha, 2021*). The assignment of stratigraphic units was traditionally biased towards bedrock in the compilation of geological maps in South Africa (*e.g.*, see the 3322 Oudtshoorn sheet: *Toerien, 1979*). The recent availability of remote-sensing datasets to assist mapping (*Doyle & Woodroffe, 2018*), a clearer understanding of onshore-offshore depositional processes influencing coastal sedimentation (*e.g.*, *Brooke et al., 2014*; *Grobler et al., 2020*) and integrated approaches in research (*Marean, Cowling & Franklin, 2020*; *Cowling et al., 2020*; *Cawthra et al., 2020b*), have greatly improved our ability to map these sediments and understand their sedimentation dynamics. The high specialization of Cape plants to the edaphic idiosyncrasies of these Cenozoic deposits (*Cowling, Holmes & Rebelo, 1992*) offers a hitherto neglected opportunity to use vegetation as an indicator of sediment age. We explore this theme here, using the highly distinctive flora and vegetation of unconsolidated, calcareous sediments of the Cape south coast as a proxy for delimitation of Holocene dunes.

Mindful of the deficiencies in the current delimitation of the coastal dune vegetation of the CFR, our primary aim for this paper is to provide a defensible scheme and map of units based on detailed floristic analysis. Our methodology for identifying, describing and mapping Cape south coast dune vegetation differs from that of previous attempts in three important ways: (1) we provide the first unified treatment of dune vegetation on the southern coast of the CFR; (2) our vegetation units are based on quantified floristic composition, with numerically defensible distinctions based on rigorous analysis; (3) our spatial delimitation of units is not reliant on geological maps. Further details on our approach are provided below. A secondary aim of the study is to interpret these vegetation patterns in terms of geographical gradients, exposure to fire, soil moisture status and sediment age. We also discuss the conservation implications of our findings.

## BIOPHYSICAL SETTING

### Geology and dune sediments

The study area extends from Cape Recife in the Eastern Cape to Cape Hangklip in the Western Cape and is nested within the Cape Floristic Region (CFR) (Fig. 1). It coincides with coastal parts of the Southeastern and Bredasdorp-Riversdale centres of endemism of the CFR. Geologically, the area incorporates the western sector of the Algoa Group and the full extent of the Bredasdorp Group, both comprising Cenozoic coastal sediments (Fig. 2; Table 1).

Coastal dunes occupy ca. 2,200 km$^2$ in the CFR, constituting just over 2% of the region (*Grobler & Cowling, 2021*). In the study area, coastal dunes cover ca. 680 km$^2$, although only ca. 620 km$^2$ of this comprises stable, vegetated dunes (Fig. 1). The largest vegetated dune systems occur in the embayments, where sediment accumulation is more favourable than the cliffed swaths of coast. These dune sediment depocentres are Walker Bay, the Wilderness-Sedgefield embayment, Oyster Bay to St Francis Bay, and Sardinia Bay to Algoa

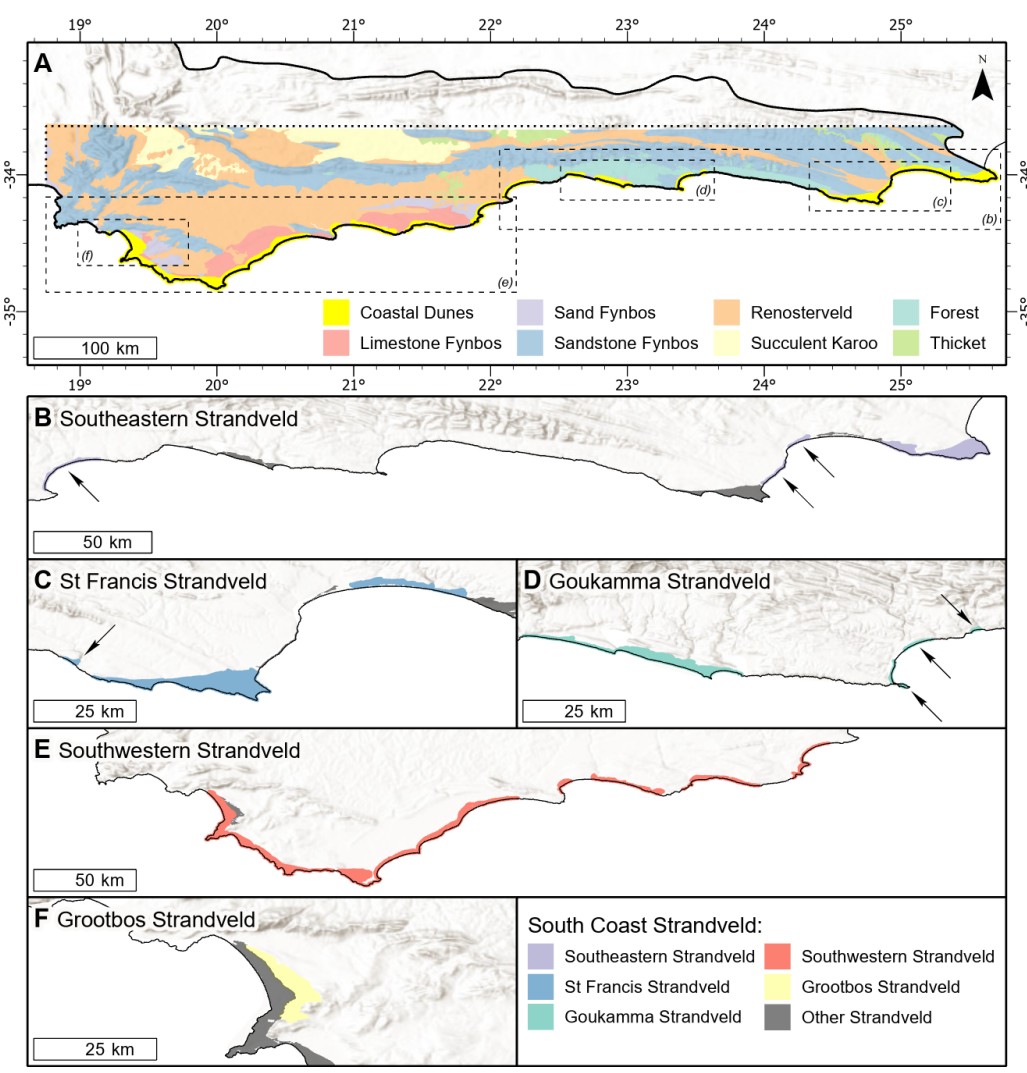

**Figure 2** **Distribution of vegetation types on the southern coastal forelands of the Cape Floristic Region.** (A) Distribution of coastal dunes (South Coast Strandveld vegetation) and vegetation types of the hinterland (after *Cowling & Heijnis, 2001*) in the study area. (B) Distribution of Southeastern Strandveld. (C) Distribution of St Francis Strandveld. (D) Distribution of Goukamma Strandveld. (E) Distribution of Southwestern Strandveld. (F) Distribution of Grootbos Strandveld. World Hillshade layer sources: ESRI, Airbus DS, USGS, NGA, NASA, CGIAR, N Robinson, NCEAS, NLS, OS, NMA, Geodatastyrelsen, Rijkswaterstaat, GSA, Geoland, FEMA, Intermap, and the GIS user community.

Bay. Generally, the Cape south coast half-heart bays coincide with underlying Bokkeveld Group shales, whereas the erosion-resistant headlands that bound them are composed of Table Mountain Group sandstones and quartzites.

The unconsolidated sediments of the Holocene dune landscapes comprise coarse to medium grained, freely draining sands with a high carbonate content (*Tinley, 1985*; *Cawthra et al., 2020b*). These attain thicknesses up to 100 m or more but are often shallower coastwards where the underlying geology—mostly Pleistocene aeolianites accreted on the basal bedrock—is often closer to the surface. Two formations are recognised: the Strandveld

**Table 1 Neogene to recent sedimentary deposits on the south coast of the Cape Floristic Region.** Available depositional ages and lithologies of correlated geological formations of the Algoa Group and Bredasdorp Group are indicated. Modified from *Roberts et al. (2006)*.

| Algoa group | Bredasdorp group | Lithology |
|---|---|---|
| Schelm Hoek Formation (undated) | Strandveld Formation Age: [onshore] 12–13 ka (*Carr, Thomas & Bateman, 2006*); 6.1–0.5 ka (*Bateman et al., 2011*); 4.1–4.7 ka; 0.8 ka (*Carr, Thomas & Bateman, 2006*); [offshore] 0.5–25.3 ka (*Cawthra et al., 2020a*) | Holocene and recently active dune fields and dune cordons |
| Nahoon Formation Age: 117 –125 ka (*Jacobs & Roberts, 2009*) | Waenhuiskrans Formation Age: 400 ka (*Roberts et al., 2012*) –36 ka (*Carr et al., 2019*); mostly MIS 5 (*Bateman et al., 2011*; *Cawthra et al., 2018*; *Helm et al., 2022*; *Roberts et al., 2008*; *Roberts et al., 2012*) | Pleistocene to Holocene calcarenite (aeolianite) with interbedded palaeosols |
| Salnova Formation (undated) | Klein Brak Formation Age: 400 ka (*Roberts et al., 2012*); MIS 5 (*Cawthra et al., 2018*) | Pleistocene estuarine coquina, calcarenite, sand and conglomerate |
| Nanaga Formation (undated) | Wankoe Formation Age: 1.7–4 Ma (*Ben Arous, Duval & Bateman, 2022*) | Pliocene aeolianite |

Formation (Bredasdorp Group) between Walker Bay and Plettenberg Bay (*Malan, 1990*), and Schelm Hoek Formation (Algoa Group) between Tsitsikamma River and Cape Recife (*Le Roux, 1989*) (Fig. 1; Table 1).

These deposits were laid down primarily since the Last Glacial Maximum (LGM) during the Postglacial Marine Transgression (*Cawthra et al., 2020a*). Most of the current extent of these sediments, however, is currently underwater on the continental shelf (*Cawthra et al., 2020b*; *Grobler et al., 2020*). The neocoastal deposits currently preserved onshore date back to ∼13 ka (*Carr, Thomas & Bateman, 2006*) but mostly started accumulating after the Holocene highstand, at ca 6 ka (*Butzer & Helgren, 1972*; *Illenberger, Rust & Vogel, 1997*; *Bateman et al., 2011*). Optically Stimulated Luminescence (OSL) dates from the seaward dune ridge at Sedgefield yield ages ranging from 6.1–0.5 ka, suggesting continuous dune building during this period (*Bateman et al., 2011*). However, coastal dunes on the Agulhas Plain postdate the highstand, having dates of 4.1 to 4.7 ka, with one site dated to 0.8 ka (*Carr, Thomas & Bateman, 2006*). Continuous dune building is evident today where most of the coastal dunefields of the Cape south coast include tracts of mobile and semi-mobile dunes, many being headland-bypass dunes that were active prior to artificial stabilization between the mid-19th and mid-20th centuries (*Tinley, 1985*; *Avis, 1989*).

In the eastern section of the study area, the Schelm Hoek Formation overlies aeolianites of Neogene age (Nanaga Formation) as well as Mid Pleistocene (Salnova Formation) and Late Pleistocene (Nahoon Formation) rocks (Table 1), the last-mentioned exposed in several sectors of the Sardinia Bay and St Francis Bay coastlines, and intermittently westwards to the Tsitsikamma River (*Le Roux, 2000*). On its landward margin, the dune cordon abuts a variably wide band of Nanaga Formation palaeodunes, except between the Kabeljous and Gamtoos Rivers, and at Jeffreys Bay where Salnova Formation gravels and Ordovician Bokkeveld Group shales, respectively, are distal to the dune cordon.

*Le Roux (2000)* mapped the aeolianites and palaeodune ridges west of St Francis Bay belonging to the Nanaga Formation, except for the mobile headland-bypass dunes that are attributed to Schelm Hoek Formation. However, based on dune morphology, vegetation and soils, we have mapped the entire dune area between St Francis Bay and Oyster Bay as Schelm Hoek Formation, as well as a band coastwards of the Oyster Bay-Tsitsikamma palaeodunes, where low and gently sloping topography have permitted Holocene dune accumulation.

To the west, Strandveld Formation dunes of the Bredasdorp Group are stacked onto Mid Pleistocene (Klein Brak Formation) cemented beach deposits, Late Pleistocene (Waenhuiskrans) aeolianites (Table 1), and on oxidised, neutral sands attributed by *Malan et al. (1994)* to the Holocene, but we suspect to be of Late Pleistocene age (these occupy the dune hinterland between the Duiwenhoks River and Vleesbaai). Klein Brak Formation rocks, comprising mainly quartz-rich, shelly sandstones, abut the dune cordon between Arniston and De Hoop Nature Reserve. Waenhuiskrans Formation sediments, some calcarenites but mostly unconsolidated, deep sands, predominate west of Cape Agulhas and in the Wilderness-Sedgefield embayment.

In the Wilderness-Sedgefield embayment, *Toerien (1979)* mapped all dune deposits as "Fixed dunes and dune rock" except for a small area of mobile dunefields that was included with Holocene dune sands ("aeolian sand"). However, guided by *Bateman et al.*'s (*2011*) OSL dates for the area, as well as dune morphology and vegetation, we have mapped the seaward dune cordons as Holocene sands, distinct from Pleistocene and Neogene aeolianites on the inland cordons. Even if only perched atop older cemented dunes on the seaward cordon, the Holocene dunes are what make up the surface geological unit in this area. On the western Agulhas Plain, *Gresse & Theron (1992)* mapped as Waenhuiskrans Formation aeolianites what we consider in terms of vegetation and dune morphology to be Strandveld Formation dune sands. Again, it may be a legacy of a mapping bias towards the cemented, rather than unconsolidated, geological material, but either way, OSL dates are required to resolve this issue.

Throughout the region, parabolic dunes with long trailing arms aligned with the prevailing wind directions (W-E) are the dominant dune morphology; running parallel to the coast, these form a series of parallel ridges ("barrier dunes") and troughs that cover the dune landscape. In the Wilderness-Sedgefield embayment and west of Oyster Bay, Holocene sands have blown onto older dune systems comprising Pleistocene and Neogene aeolianites, respectively (*Illenberger, Rust & Vogel, 1997*; *Bateman et al., 2011*). *Tinley (1985)* calls these wind rift dunes. Hummock-blowouts and playa lunette dunes are uniquely found in the high wind energy environment west of Cape Agulhas (*Tinley, 1985*). More detail on dune morphology is given in the vegetation descriptions below.

## Dune soils

The soils of Holocene coastal dunes are relatively deep, except in the few places where calcarenites, cemented duricrusts and Table Mountain Group rocks outcrop. The soils are alkaline (Ph 7-8), and well- to excessively-drained, coarse- to medium-grained sands with high content of shell material (*Cowling, 1984*; *Tinley, 1985*; *Cawthra et al., 2020b*).

Phosphate levels are high (10-40 ppm available P) in comparison to most other CFR soils, and base saturation is also high, driven mainly by calcium carbonate. Organic carbon and total nitrogen is considerably higher under thicket than fynbos vegetation, owing to the higher input of litter and higher shade beneath the thicket canopy (*Cowling, 1984*; *Tinley, 1985*).

Coastal dune soils differ markedly from those of the Pleistocene and Neogene coastal sediments exposed landward of the Holocene cordon (*Tinley, 1985*; *Cawthra et al., 2020b*). These older dune substrates comprise two main groups: skeletal sands (0–0.3 m) on calcarenites, and deep (>1.0 m), unconsolidated sands. The skeletal sands, like those of coastal dunes, are calcareous, alkaline and relatively rich in P. They differ in that they are everywhere shallow. The unconsolidated sands differ from coastal dune sands in that they are more weathered, finer grained, and with better water holding capacity. They differ further in being neutral to weakly acid and having lower levels of all soil nutrients. As we detail below, the vegetation on the older sediments is different floristically and structurally from coastal dune vegetation, there being very high to complete turnover (beta diversity) of species across the edaphic boundary (*Cowling, 1990*; *Cowling & Holmes, 1992*).

## Climate

The climate varies from winter-rainfall (Cape Hangklip to Arniston) to a bimodal regime for the remainder of the study area that shows peaks in spring and autumn (Fig. 1) (*Schulze, 2008*; *Bradshaw & Cowling, 2014*). While the proportion of warm-season rainfall (summer –autumn) increases eastwards, this period corresponds with the highest drought stress for plants throughout the region. Thus, the rainfall regime can be characterised as Mediterranean to sub-Mediterranean. Mean annual rainfall for Holocene dune cordons is highest in the Wilderness-Sedgefield area (ca. 800 mm at Goukamma) and between Cape St Francis and the eastern Tsitsikamma (700–900 mm). Elsewhere, it is lower: 550–600 mm between Cape Recife (Gqeberha) and the Gamtoos River, and <500 mm between the Groot Brak River and Danger Point. Temperature regimes show little annual variation: mean midsummer temperatures range from 18 to 22 °C and mean midwinter temperatures from 14 to 16 °C, and the region is frost free. Wind regimes are fierce with numerous strong- to gale force winds in summer (from the east) and in winter (from the west). The central region, between Mossel Bay and Tsitsikamma, has a calmer wind regime (*Schulze, 2008*).

## Flora & vegetation

This region forms part of the CFR, the second richest plant biodiversity hotspot globally (*Grobler & Cowling, 2022*). Coastal dune floras of the region are rich, despite the low ecological heterogeneity of dune landscapes as well as their small areas (*Grobler et al., 2020*). Indeed, they are as rich in species as some floras of the coastal hinterland that are associated with ancient sediments, and considerably richer than dune floras from other Mediterranean Climate Regions, except for some sites in the Mediterranean Basin (*Grobler et al., 2020*). Like Holocene dune floras of other Mediterranean Climate Regions, dominant families on CFR dunes are Asteraceae, Poaceae and Fabaceae; the high incidence of species belonging to Iridaceae, Aizoaceae and Scrophulariaceae is a distinctive feature

of Cape dune floras (*Grobler & Cowling, 2021*). Almost half the species are endemic to the Greater Cape Floristic Region (CFR plus the Succulent Karoo), and ca. 40% are endemic to the CFR. Species endemic to calcareous substrata (dunes and calcarenites) comprise up to 30% of local floras and are overwhelmingly of Cape affinity. Most are dwarf shrubs and geophytes. *Grobler et al. (2020)* have attributed the surprisingly high diversity and endemism of the floras south Cape Holocene dunes—relative to the small and fragmented contemporary area of dune landscape—to the extensive distribution of coastal dunes on the Palaeo-Agulhas Plain that was variously exposed during the glacial periods that prevailed during the Pleistocene.

The major biomes associated with Cape Holocene coastal dunes are Fynbos and Subtropical Thicket; there are also minor occurrences of Grassland (in the east), Forest and Wetland biomes (*Cowling et al., 2019*; *Grobler & Cowling, 2021*). Most species (40–60%) in local floras are associated with the Fynbos Biome, followed by the Subtropical Thicket biome (10–20%). The vegetation of Holocene coastal dunes comprises a mosaic of open-canopy fynbos and closed-canopy subtropical thicket. The former is dominated by ericoid shrubs and restioids of Cape affinity and prevails in fire-exposed and edaphically dry sites; the latter is dominated by broad-leaved shrubs and low trees of tropical affinity and grows in moist and fire-protected sites where it can attain forest stature (*Cowling, 1984*; *Tinley, 1985*; *Rebelo et al., 1991*; *Cowling et al., 1997*). Biomass of dune fynbos is dominated by species endemic to Holocene dunes, although non-endemics comprise most of the species growing there (*Cowling et al., 2019*). Dune thicket, comprising lineages of tropical origin, is a distinctive formation that includes several species endemic to the coastal dunes of the CFR (*Cowling, 1983*; *Vlok, Euston-Brown & Cowling, 2003*).

Like fynbos of inland habitats (*Kraaij & Van Wilgen, 2014*), the fynbos component of southern Cape dune vegetation is highly fire-prone and comprises numerous non-sprouting species that recruit from seed only after fire (*Cowling & Pierce, 1988*; *Pierce & Cowling, 1991a*; *Pierce & Cowling, 1991b*; *Cowling et al., 2019*). While dune thicket (and forest) is best developed in fire-free sites, most component species, especially dune endemics, resprout readily after fire (*Cowling et al., 2019*; *Strydom et al., 2020*) and are resilient to even high-intensity burns (*Strydom et al., 2020*; *Strydom et al., 2023*).

## MATERIALS & METHODS

In this study, we employed a phytosociological approach to identify and describe the different vegetation units that occur on coastal dunes of the Cape south coast and to inform mapping their distributions in the study area. Recent attempts at delimiting these plant associations (*Mucina et al., 2006a*; *Mucina et al., 2006b*; *Rebelo et al., 2006*; *Grobler et al., 2018*) did not include formal floristic analyses, resulting in disparate classification at the biome level and inappropriate combination/splitting of vegetation units across/within edaphic substrata and physiographic units. While *Rebelo et al.*'s (*2006*) treatment of Fynbos Biome vegetation recognised the Walker Bay to Oyster Bay part of our study area as belonging to a coherent unit—South Strandveld—this treatment excluded coastal dunes occurring eastward to Cape Recife that share several characteristic species with
the remainder of this region (*Cowling, 1983*; *Grobler & Cowling, 2021*). Similarly, *Grobler et al.*'s (*2018*; see also *Vlok, Euston-Brown & Cowling, 2003*) treatment of coastal dune vegetation considered only that stretch of coast between the Duiwenhoks River and Cape Recife and excluded the coastal dunes westward to Cape Hangklip. This currently fragmented treatment of southern Cape coastal dune vegetation is further marred by inherent methodological differences: circumscription of Fynbos Biome vegetation units ("Western Strandveld"; *Rebelo et al., 2006*), for example, relied heavily on extant, coarse-scale geological maps (1:250 000) that are now understood to be inadequate, especially maps of coastal sediments (*Botha, 2021*); Thicket Biome units ("Dune Thicket"; *Grobler et al., 2018*; *Vlok, Euston-Brown & Cowling, 2003*), on the other hand, were delimited without explicit consideration of edaphic substrata and associated turnover in plant communities (*e.g.*, *Cowling, 1980*; *Cowling & Holmes, 1992*; *Thwaites & Cowling, 1988*).

Phytosociological studies provide major advantages for classifying vegetation over an approach based on expert opinion and geology, as was used by *Rebelo et al. (2006)* for mapping and describing the vegetation of Cape coastal dunes. Firstly, these studies offer a more objective and systematic approach to vegetation classification by relying on quantitative data and statistical analyses, thus reducing biases and inconsistencies that may arise from relying on individual expert judgments. Secondly, phytosociological studies provide a more comprehensive and detailed description of vegetation communities by undertaking a thorough inventory of species and their relative abundances, thus facilitating more accurate and precise classification. Thirdly, phytosociological studies can provide a more repeatable and standardized approach to vegetation classification, allowing for consistent application of methods and criteria across different sites and regions, and facilitating comparisons and generalizations across studies (*Chytrý, Schaminée & Schwabe, 2011*). Overall, phytosociological studies offer a rigorous and scientific approach to vegetation classification that can yield more reliable and informative results compared to expert-opinion approaches.

## Data collection

Our data on floristic composition of coastal dune vegetation stems from two primary sources, namely: (1) relevé data collected as part of previous phytosociological studies in the Cape south coast that included vegetated coastal dunes; and (2) relevé data collected by us during a recent vegetation survey of the study area (during the austral spring to early summer season of 2018 and 2019).

From the literature we were able to source relevés for dune vegetation at Cape St Francis (*Cowling, 1984*), Goukamma near Knysna (*Van der Merwe, 1976*) and Grootbos near Gansbaai (*Mergili & Privett, 2008*). We excluded any relevés from azonal vegetation associated with the coastal zone (*e.g.*, mobile dune sheets, foredunes, nearshore herblands and shrublands under strong maritime influence) and with poorly drained sites (*e.g.*, various types of wetlands), or those from vegetation shaped by anthropogenic disturbances (*e.g.*, plots with abundant alien invasive species, vegetation subjected to intense grazing or mowing). We also excluded relevés from older Pleistocene and Neogene coastal sediments. Importantly, relevés from both thicket and fynbos vegetation were included,

thus incorporating samples across the full gamut of the dune vegetation mosaic. A total of 195 relevés were included from these sources, 135 sampled from dune fynbos and 60 sampled from dune thicket (Fig. 1A).

After collating the above data, we identified seven dune areas along the Cape south coast—an area that has hosted relatively few phytosociological studies (*Rutherford, Mucina & Powrie, 2012*)—for which floristic data were conspicuously lacking. These included the dune cordons at the following locations: Cape Recife south of Gqeberha; between the Gamtoos and Kabeljous rivers; Hartenbos east of Mossel Bay; Vleesbaai west of Mossel Bay; Geelkrans near Still Bay; Arniston east of Cape Agulhas; and Brandfontein west of Cape Agulhas (Fig. 1A). Approval to access the dune cordon near Arniston was granted by the Denel Overberg Test Range. These areas represent samples of dune cordons of various sizes and configurations, but all include moderate to large areas of stable, vegetated dunes. In line with the generally accepted protocol for phytosociological classification and description of vegetation in South Africa (*Brown et al., 2013*; *Brown & Bredenkamp, 2018*), we conducted vegetation surveys at each of these sites following the Zürich-Montpellier (Braun-Blanquet) approach (*Werger, 1974*; *Westhoff & Van der Maarel, 1978*). Stratified random placement of sample plots was employed to ensure that floristically and environmentally homogeneous stands of vegetation were sampled, with plots primarily stratified between stands of the fynbos and thicket cover states. Plots were located in vegetation without recent disturbances, although this was not possible for Cape Recife as the entire dune landscape burned during two successive wildfires in 2018 and 2019. For fynbos, we sampled plots of $5 \times 5$ m ($25$ m$^2$), and $10 \times 10$ m ($100$ m$^2$) for thicket (*Brown et al., 2013*). Within each plot, the total vascular plant flora was recorded and cover-abundance values, using the modified Braun-Blanquet scale (*Kent, 2012*), were estimated for each species. We sampled a total of 58 plots during our survey, including 45 in fynbos and 13 in thicket (Fig. 1A).

Our final dataset, combining vegetation plot data from our field survey with those gleaned from the literature, comprised 253 relevés (180 fynbos, 73 thicket) from 10 coastal dune sites (Data S1). In total, 500 plant species—representing about 50% of the coastal dune flora in the study area (*Grobler & Cowling, 2021*)—were recorded, with nomenclature following the Botanical Database of Southern Africa (*SANBI, 2022*).

## Data analysis

We used non-metric multidimensional scaling (NMDS), implemented with the 'metaMDS' function of the vegan *v.* 2.6.2 (*Oksanen et al., 2022*) R *v.* 4.2.1 statistical software (*R Core Team, 2022*) package, to ordinate our vegetation samples. Prior to ordination, data were subjected to Wisconsin double standardization (scaled to zero mean and unit variance) and square-root transformation. The NMDS used Bray-Curtis dissimilarity as a distance metric along two dimensions ($k = 2$). We then used standard procedures (*Kent, 2012*; *Oksanen et al., 2022*), as previously described in *Strydom et al. (2022)*, to complete the ordination analysis. Specifically, the ordination was run 999 times with random starts to prevent the NMDS from becoming trapped in local optima, and the solution with minimal stress was then selected. To facilitate interpretation of the results, the final NMDS solution was centred and rotated by principal components so that the variance of points

was maximised along the first NMDS axis, which allows inference about major gradients between ordinated samples. The resulting ordination plots were examined to identify any clustering or separation of vegetation samples.

To facilitate classification of our vegetation samples, we used the juice JUICE *v.* 7.1.30 software (*Tichý, 2002*) to conduct two-way indicator species analysis (TWINSPAN) and to construct synoptic tables of our vegetation units. We used the modified TWINSPAN algorithm developed by *Roleček et al. (2009)* with pseudospecies cut levels of 0, 5, 15, 25, 50 and 75 for divisive clustering of the whole dataset. This, together with exploratory NMDS analyses, showed that samples were clearly divided between two major groups—fynbos and thicket. We thus used separate datasets for each of these groups for subsequent multivariate analyses. Two-way phytosociological tables produced by the TWINSPAN classifications were further refined based on our ordinations of the vegetation groups and following standard Braun-Blanquet procedures (*Werger, 1974*). Synoptic tables indicating species constancy, fidelity and average cover were produced for our final vegetation units (Table S1). Additionally, we calculated diversity metrics ($\alpha$ and $\beta$) following *Whittaker (1972)* and proportional edaphic endemism (*i.e.,* dune endemics) of component species for each vegetation unit, with edaphic endemism following *Grobler & Cowling (2021)*.

## Vegetation mapping

We mapped our vegetation units in the ESRI ArcGIS Pro *v.* 2.5.0 software. The initial mapping process relied on recent (captured since 28 February 2021), high-resolution (0.05–0.60-m resolution with 5.0-m accuracy) Maxar Vivid™ satellite imagery of the study area available from the ESRI ArcGIS Online 'World Imagery' layer (https://www.arcgis.com/home/item.html?id=10df2279f9684e4a9f6a7f08febac2a9). In cases where the dune topography was less distinct and where transitions between adjacent edaphic substrates and associated vegetation were not visually evident, the boundaries of dune vegetation types were aligned with those mapped in other vegetation mapping exercises (*Mergili & Privett, 2008*). In areas where we had incomplete knowledge of vegetation boundaries, we used our criteria for identifying Holocene dunes (see above) as a surrogate for mapping vegetation associated with these substrata. Forested areas identified in the latest VEGMAP (*Dayaram et al., 2019*) were excluded, in particular patches of Southern Coastal Forest (FOz 6; *Mucina et al., 2006a*; *Mucina et al., 2006b*) occurring in dunes of the far western sector of our study area. Note that unmapped patches of dune forest, which should arguably be included in the Southern Coastal Forest type, also occur in the far east of our study area in dune fields between Wilderness and Gqeberha (*Cowling et al., 2019*; *Strydom et al., 2022*). Similar to mapped forests, we aligned our vegetation boundaries with those of non-terrestrial coastal ecosystems (*e.g.,* estuaries, shores) (*Harris et al., 2019*). Where our boundaries did not differ substantially from those in the VEGMAP and where Holocene dune topography was not unambiguously evident in satellite imagery, we aligned vegetation type boundaries to reduce the need for extensive post-processing, thereby reducing the potential introduction of mapping and topology errors. Spatial data produced by our mapping exercise are provided in Data S2.

## RESULTS & DISCUSSION

### Delimitation and characterisation of the Strandveld concept

In characterising Holocene age dune vegetation of the Cape south coast, we have invoked the Strandveld concept (*Acocks, 1953*). Strandveld (literally "beach vegetation") is a term embedded in the Cape Afrikaans vernacular dating back to the Dutch settlement of the Cape west and south coasts, starting in the 17th Century. An alternative term is Duineveld ("dune vegetation"). As it appears on maps and from published accounts, Strandveld refers to areas of coastal, calcareous, white ("wit") sands along the western part of the south Cape coast, and the beige sands of the eastern part. It is distinct from Sandveld ("sand vegetation") which refers to the vegetation on coastal Pleistocene sands immediately inland of the Strandveld, where the neutral to acid sands are mostly weathered yellow to reddish brown and the vegetation is invariably Sand Fynbos (*Thwaites & Cowling, 1988*; *Rebelo et al., 2006*; *Cawthra et al., 2020b*). We suggest using Strandveld to denote the vegetation of the Holocene calcareous sands of the Cape coast. Strandveld has poorly developed, quick draining soils with high levels of exchangeable cations and relatively high levels of P and N; these support a vegetation dominated by shrubs comprising mosaics and mixtures of fynbos and thicket species. The older sands supporting Sand Fynbos are more weathered, finer-grained, more water retentive and less fertile than the Strandveld sands. While Sand Fynbos west of the Breede River mouth has a minor thicket component, this is not the case to the east where dune thicket with forest patches in moist, fire-protected sites, forms an important component of the vegetation (Figs. S1A–S1B).

Thus defined, our Strandveld concept excludes the vegetation termed Strandveld by *Rebelo et al. (2006)* that is associated with granite, calcarenite and Pleistocene sands on the west coast but includes the vegetation on Holocene sands (Witsand Formation of the Sandveld Group). Being floristically dominated by fynbos taxa, and thus belonging to the Fynbos Biome, the concept also excludes vegetation of the Holocene dunes north of Elands Bay (where succulent karoo taxa replace fynbos lineages) (*Rebelo et al., 2006*) and east of Algoa Bay (where thicket prevails and where depauperate fynbos is confined to ecologically demanding sites, such as dune slipfaces and excessively-drained ridges) (*Hoare et al., 2006*; *Taylor & Morris, 1981*). This vegetation is best placed in the Succulent Karoo and Subtropical Thicket biomes, respectively. The vegetation on coastal cliffs composed of granites, sandstones and cemented aeolianites, that are mapped as Strandveld by *Rebelo et al. (2006)*, are also excluded from our Strandveld concept. These sites, which are under strong maritime influence and support a wind-pruned vegetation comprising a diverse array of species tolerant of exposure to salt-laden winds (*Taylor & Boucher, 1993*), are best included in the azonal Cape Seashore Vegetation concept, a heterogeneous assemblage of soft- and hard-substratum communities (*Mucina et al., 2006a*; *Mucina et al., 2006b*).

Amongst Fynbos Biome vegetation types, Strandveld has the highest cover of Subtropical Thicket Biome species, leading to some suggesting that it should be included in the latter biome (*Low & Rebelo, 1998*; *Vlok, Euston-Brown & Cowling, 2003*; *Grobler et al., 2018*). However, while closed-canopy clumps of thicket may be locally dominant in Strandveld, the most extensive vegetation is fynbos, albeit with an admixture of thicket species;

moreover, Strandveld floras are dominated by Cape taxa (*Cowling et al., 2019*; *Grobler & Cowling, 2021*).

## South coast strandveld

Our focus is on South Coast Strandveld, vegetation associated with poorly weathered, calcareous sands of Holocene age that comprises a matrix of closed-canopy thicket and asteraceous (ericoid shrub-dominated) fynbos (*Cowling et al., 1988*) that occurs along the Cape south coast. It corresponds to *Rutherford, Mucina & Powrie*'s (*2006*) South Coast Strandveld Bioregion of the Fynbos Biome and to the south coastal vegetation units of *Rebelo et al.*'s (*2006*) Western Strandveld intrazonal vegetation group. Also included by us in South Coast Strandveld is the Holocene dune vegetation between Oyster Bay and Cape Recife, which *Mucina et al.*'s (*2006a*) and *Mucina et al.*'s (*2006b*) map as Algoa Dune Strandveld, part of their azonal Eastern Strandveld vegetation group. Following *Moll et al. (1984)* and *Rutherford, Mucina & Powrie (2006)*, we propose two strandveld bioregions for the Cape Floristic Region: West Coast Strandveld and South Coast Strandveld. Justification for this delimitation is provided later in this section.

While the fynbos component of South Coast Strandveld often includes low-growing thicket taxa, most of which are Strandveld endemics (*e.g.*, *Robsonodendron maritimum*, *Rapanea gilliana*), our data shows that the pockets of unbroken thicket are floristically well differentiated from fynbos (Fig. 3). Moreover, thicket differs structurally, functionally and phylogenetically from fynbos (*Cowling, 1983*; *Cowling et al., 1997*).

South Coast Strandveld is characterised by the presence of *Olea exasperata* and *Restio eleocharis* (Fig. 4), both largely restricted to calcareous, Holocene dunes. Other species widely distributed in South Coast Strandveld and with the bulk of their populations in this unit, are *Searsia crenata*, *Cussonia thyrsiflora*, *Restio leptoclados* and *Metalasia muricata*.

As such, South Coast Strandveld includes *Rebelo et al.*'s (*2006*) Overberg Dune Strandveld (Fs 7), Blombos Strandveld (Fs 8), Groot Brak Dune Strandveld (Fs 9), Southern Cape Dune Fynbos (FF d 11) and *Mucina et al.*'s (*2006a*) and *Mucina et al.*'s (*2006b*) Algoa Dune Strandveld (AZ s1) (Table 2). It excludes *Rebelo et al.*'s (*2006*) Langebaan Dune Strandveld (FS 5) and Cape Flats Dune Strandveld (FS 6), which form part of *Rutherford, Mucina & Powrie*'s (*2006*) West Strandveld Bioregion. In both these units, *Restio eleocharis* and *Olea exasperata* are rare (*Boucher & Jarman, 1977*), while thicket species common to these units—*Maytenus undata* and *Maurocentia frangula*—are encountered only in the extreme west of South Coast Strandveld (*Boucher, 1978*). Following *Moll et al. (1984)*, we suggest these units form West Coast Strandveld, which includes many species largely restricted to the strongly winter-rainfall west coast, notably amongst dwarf shrubs (*e.g.*, *Didelta carnosa*, *Limonium perigrinum*), succulent shrubs (*e.g.*, *Ruschia geminiflora*, *Pelargonim gibbosum*), graminoids (*e.g.*, *Cladoraphus cyperoides*, *Thamnochortus spicigerus*) and annuals (*Zaluzianskya* spp., *Grielum grandiflorum*) (*Rebelo et al., 2006*). However, in many respects, Overberg Dune Strandveld, especially west of Gansbaai, is transitional

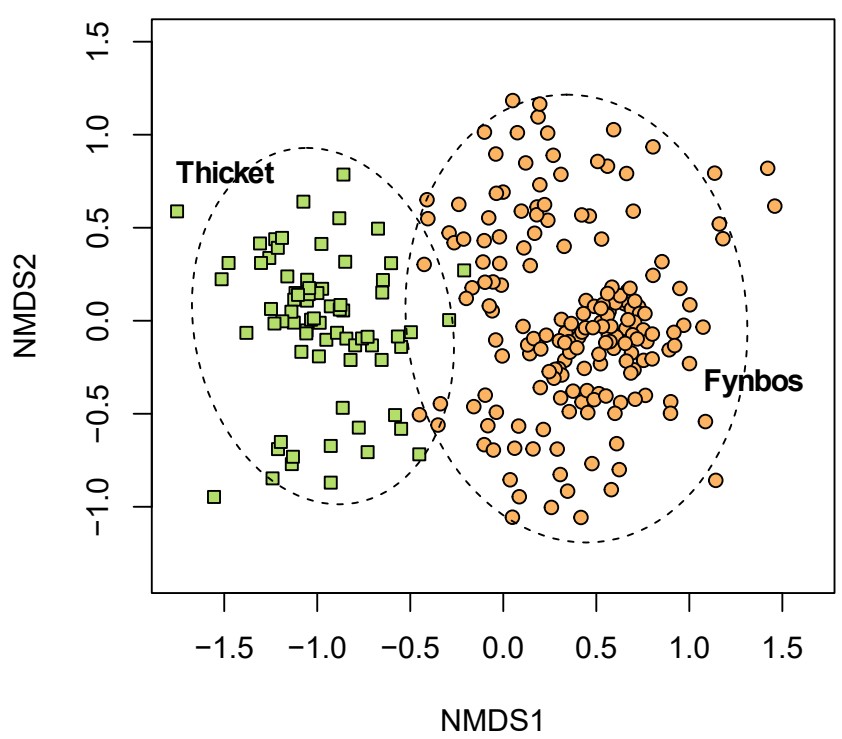

**Figure 3** **Non-metric multidimensional scaling (NMDS) of the fynbos and thicket plots sampled in this study.** NMDS solution: $k = 2$; stress = 0.206; non-metric $R^2 = 0.958$; linear $R^2 = 0.811$.

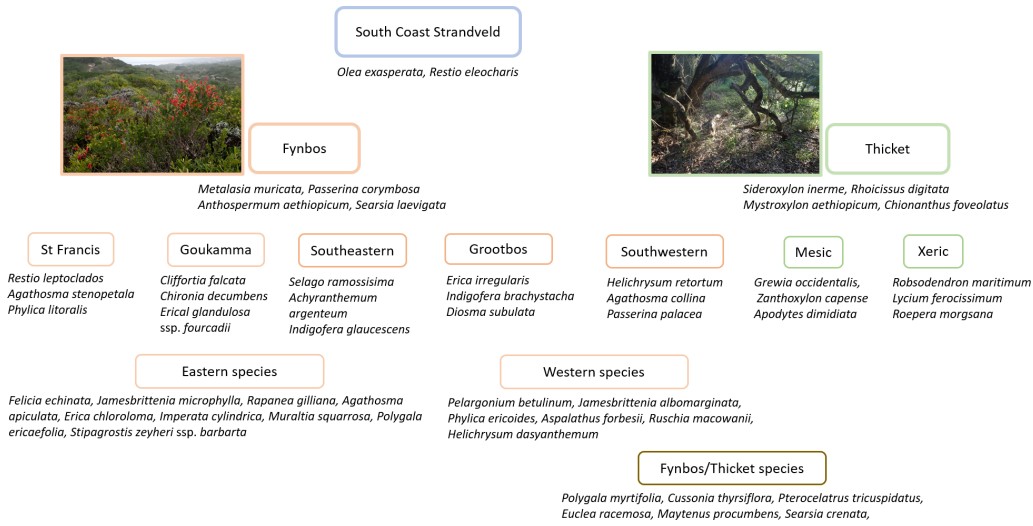

South Coast Strandveld

*Olea exasperata, Restio eleocharis*

Fynbos

*Metalasia muricata, Passerina corymbosa
Anthospermum aethiopicum, Searsia laevigata*

Thicket

*Sideroxylon inerme, Rhoicissus digitata
Mystroxylon aethiopicum, Chionanthus foveolatus*

St Francis

*Restio leptoclados
Agathosma stenopetala
Phylica litoralis*

Goukamma

*Cliffortia falcata
Chironia decumbens
Erical glandulosa
ssp. fourcadii*

Southeastern

*Selago ramossisima
Achyranthemum
argenteum
Indigofera glaucescens*

Grootbos

*Erica irregularis
Indigofera brachystacha
Diosma subulata*

Southwestern

*Helichrysum retortum
Agathosma collina
Passerina palacea*

Mesic

*Grewia occidentalis,
Zanthoxylon capense
Apodytes dimidiata*

Xeric

*Robsodendron maritimum
Lycium ferocissimum
Roepera morgsana*

Eastern species

*Felicia echinata, Jamesbrittenia microphylla, Rapanea gilliana, Agathosma
apiculata, Erica chloroloma, Imperata cylindrica, Muraltia squarrosa, Polygala
ericaefolia, Stipagrostis zeyheri ssp. barbarta*

Western species

*Pelargonium betulinum, Jamesbrittenia albomarginata,
Phylica ericoides, Aspalathus forbesii, Ruschia macowanii,
Helichrysum dasyanthemum*

Fynbos/Thicket species

*Polygala myrtifolia, Cussonia thyrsiflora, Pterocelatrus tricuspidatus,
Euclea racemosa, Maytenus procumbens, Searsia crenata,*

**Figure 4** **Diagnostic species for Strandveld vegetation types recognized in this study.**

**Table 2 Correspondence between South Coast Strandveld units described in this study and corresponding National Vegetation Map (VEGMAP) units.** VEGMAP units were described by *Mucina et al. (2006a)*, *Mucina et al. (2006b)*, *Rebelo et al. (2006)* and *Grobler et al. (2018)*.

| VEGMAP 2006 | Notes | VEGMAP 2018 | Notes | This study |
|---|---|---|---|---|
| Overberg Dune Strandveld (Fynbos Biome) | Grootbos Strandveld not recognised as a separate unit | Overberg Dune Strandveld (Fynbos Biome) | Grootbos Strandveld not recognised as a separate unit | Southwestern Strandveld Grootbos Strandveld |
| Blombos Strandveld (Fynbos Biome) | Coincides with the eastern part of Southwestern Strandveld | Blombos Strandveld (Fynbos Biome) Hartenbos Dune Thicket (Subtropical Thicket Biome) | All mapped as Hartenbos Dune Thicket (Subtropical Thicket Biome) except for a sliver between Duiwenhoks and Breede rivers; most Hartenbos Dune Thicket on Mid Pleistocene aeolianites | Southwestern Strandveld |
| Groot Brak Dune Strandveld (Fynbos Biome) | Coincides with the western part of Southeastern Strarndveld | Groot Brak Dune Strandveld (Fynbos Biome) | Minor coincidence with Strandveld; most mapped on Mid Pleistocene aeolianites and other substrata | Southeastern Strandveld |
| Southern Cape Dune Fynbos (Fynbos Biome) | Classified as a form of Sand Fynbos associated with neutral to acid sands on Pleistocene aeolianites; In Cape St Francis area, mobile (late Holocene) dunefields are mapped as this type whereas the vegetation on earlier Holocene aeolianites are mapped as Algoa Dune Strandveld | Goukamma Dune Thicket and St Francis Dune Thicket (Subtropical Thicket Biome) | Mapped area extends on to Mid- Pleistocene and Neogene aeolianites | Goukamma Strandveld St Francis Strandveld (in part) |
| Algoa Dune Strandveld (Azonal Coastal Biome) | Classified as a form of Eastern Strandveld, an azonal coastal vegetation | St Francis Dune Thicket (Subtropical Thicket Biome) | Extends beyond Cape Recife to include thicket on Holocene Dunes and Neogene aeolianites | St Francis Strandveld Southeastern Strandveld |

between Strandveld of the south and west coasts. More quantitative research is required to resolve the boundary between South Coast and West Coast Strandveld.

A recent revision of the *Mucina & Rutherford (2006c)* vegetation map of South Africa to incorporate Subtropical Thicket biome concepts (*Grobler et al., 2018*) is not helpful regarding the delimitation of South Coast Strandveld as defined here. A major problem is that this revised system attributes what we define as Strandveld to the Subtropical Thicket biome when, as we have mentioned above, Strandveld is a Fynbos Biome vegetation. Also, many of these revised vegetation units occur on both Holocene and Pleistocene sediments. Thus, much of what we map as Southwestern and Southeastern Strandveld between the Groot Brak and Duiwenhoks rivers is included in *Grobler et al.*'s (*2018*) Hartenbos Dune Thicket, a Subtropical Thicket biome vegetation that extends on to Pleistocene sediments (Table 2). What we map as Goukamma Strandveld in the Wilderness-Sedgefield embayment is included in Goukamma Dune Thicket, which also extends on to Pleistocene sediments. The entire dunefield between the Oyster Bay and Kromme River and all Holocene dunes eastwards to Algoa Bay are mapped by *Grobler et al. (2018)* as St Francis Dune Thicket.
We include the former in St Francis Strandveld, and the bulk of the latter in Southeastern Strandveld (Kromme River to Cape Recife). Overall, as pointed out above, we disagree that the vegetation of Holocene dunes in the south eastern part of the CFR (*i.e.,* between Cape Recife and the Duiwenhoks River) should be included in the Subtropical Thicket Biome (*Cowling et al., 2019*; *Grobler & Cowling, 2021*).

The boundary between South Coast Strandveld and the adjacent Sand Fynbos and Limestone Fynbos units (Fig. S1) requires some discussion, since some species are shared amongst these three vegetation units (*Grobler & Cowling, 2021*). *Rebelo et al. (2006)* suggest that higher fire return intervals in Sand Fynbos are of prime importance in floristically and structurally differentiating it from Strandveld, with soil nutrients playing a lesser role. They do point out, that "[W]ith time, the alkaline sands [supporting Strandveld] become leached and are invaded by sand fynbos…". While there is no evidence in support of the fire return interval hypothesis, there is good evidence that Sand Fynbos occupies soils that are more acidic and of lower nutrient status than Strandveld (*Cawthra et al., 2020b*), a consequence of nutrients leaching from the older sediments. Moreover, Strandveld thicket species are fire-resilient, resprouting vigorously after fire (*Strydom et al., 2020*), even at high fire return intervals (*Strydom et al., 2023*).

An alternative hypothesis is that the higher incidence of thicket in Strandveld relative to Sand Fynbos may be a consequence of higher water tables owing to the impervious bedrock approaching the surface in these coastal settings. These conditions would favour thicket over fynbos, especially in dune swales and the immediate coastline, where water tables are near the surface (*Cowling, 1984*; *Cowling & Hoffman, 2021*). Where the water table is deep, such as inland settings and on the crests and upper slopes of high dune ridges, thicket incidence is much lower. However, more research is required to test this soil moisture hypothesis. In a later section we speculate how soil moisture and fire exposure interact to determine the coexistence of thicket and fynbos cover states in South Coast Strandveld.

Limestone Fynbos is easily differentiated edaphically from Strandveld based on the former being confined to skeletal sands overlying calcarenite as opposed to the deep unconsolidated sands of Strandveld (*Thwaites & Cowling, 1988*). Differences in soil nutrients between the two types are slight (*Thwaites & Cowling, 1988*; *Cawthra et al., 2020b*).

## Fynbos biome vegetation

As pointed out above, the flora of south coast Holocene dunes is dominated by species associated with the Fynbos Biome, and these provide the basis for delimiting the five Strandveld units we recognise in this study. Owing to the more widespread occurrence of their component species, dune thicket communities extend over wider geographical ranges (see below) and are embedded in the Strandveld units (hence the term "dune fynbos-thicket mosaic" used in earlier accounts) (*Cowling, 1984*; *Cowling & Heijnis, 2001*). Below, we briefly describe each of these five fynbos-dominated units found on the Cape south coast.

### Southeastern Strandveld

Southeastern Strandveld (Fig. 5A) occurs on the southeastern and northeastern sectors of the St Francis Bay coastline as well as the dunes between the Groot Brak River and Mossel Bay (Fig. 2). It thus includes part of *Grobler et al.*'s (*2018*) St Francis Dune Thicket and Hartenbos Dune Thicket, as well as *Rebelo et al.*'s (*2006*) Groot Brak Dune Strandveld (Tables 2, 3). In total, Southeastern Strandveld covers an area of ca. 119 km². It is poorly differentiated from the other strandveld types showing strongest floristic links to Southwestern Strandveld and St Francis Strandveld (Fig. 6). The western enclave near Mossel Bay lacks several species prominent in the eastern sector.

Other than the relatively broad dune-field between Sardinia Bay and Cape Recife, the dune cordon supporting this unit is narrow, in many places not exceeding 500 m in width. These Holocene dunes (Schelm Hoek Formation in the east and Strandveld Formation in the west) mostly comprise parallel ridges of hairpin or parabolic types. Those sourced at Sardinia Bay included mobile headland bypass dunes, transporting coastal sediment from St Francis Bay to Algoa Bay, prior to stabilization in the latter part of the 19th Century (*Avis, 1989*). This band of dunes is underlain by Late Pleistocene aeolianites (Nahoon Formation) and Table Mountain Group sediments. On its inland margin, the Holocene dunes abut Neogene sands of the Nanaga Formation (east) and sands of Wankoe Formation (west). These sediments support a Sand Fynbos, grassy in the east and restioid in the west.

The climate of Southeastern Strandveld is warm temperate, subhumid to semi-arid and sub-Mediterranean. Annual rainfall is highest in the extreme east at Cape Recife (550–600 mmyr⁻¹) and lower elsewhere (500 mmyr⁻¹) (Fig. 1). The mid-summer months (Dec–Feb) are the driest and the time when plant moisture stress is most severe. Rainfall reliability is highest in the equinoctial months. The proportion of warm season rainfall decreases from east to west. The temperature regime is equable: mean midsummer temperatures are ca 20−22 °C, and midwinter temperatures ca. 16−18 °C. The coast is everywhere windy but notably so at Cape Recife. West-southwest winds prevail in winter and east-southeast winds in summer.

Frequently encountered but not dominant shrub species that differentiate Southeastern Strandveld include *Crassula muscosa*, *Selago ramosissima*, *Achyranthemum argenteum*, *Indigofera glaucescens* and *I. verrucosa* (Fig. 4, Table S1) Less frequently encountered differential shrubs are *Helichrysum versicolor*, *Otholobium* sp. nov. 'algoensis', *Indigofera porrecta* and *Pharnaceum thunbergii*. Dominant shrubs that are best represented in this unit include *Muraltia spinosa*, *Coleonema pulchellum*, *Chrysocoma ciliata* and *Agathosma apiculata*. Xeric Dune Thicket shrubs are well represented in Southeastern Strandveld which, especially along narrow dune cordons, may dominate the landscape. Species characteristic of this unit include *Aloe africana*, *Rapanea gilliana*, *Dovyalis rotundifolia* (St Francis Bay coast) and *Aloe arborescens* (Mossel Bay coast) (Fig. 6A). The xeric nature of the vegetation is reflected in the high occurrence and cover of succulents; in addition to those already mentioned above, are *Crassula nudicaulis*, *C. expansa* ssp. *filicaulis*, *Carpobrotus deliciosus*, *Mesembryanthemum aitonis* and *M. canaliculata*.

With 104 species sampled in only 11 plots, and a $\beta$-diversity score of 9.45 (Table 4), South Eastern Strandveld has a rich flora characterised by high turnover from site to site.

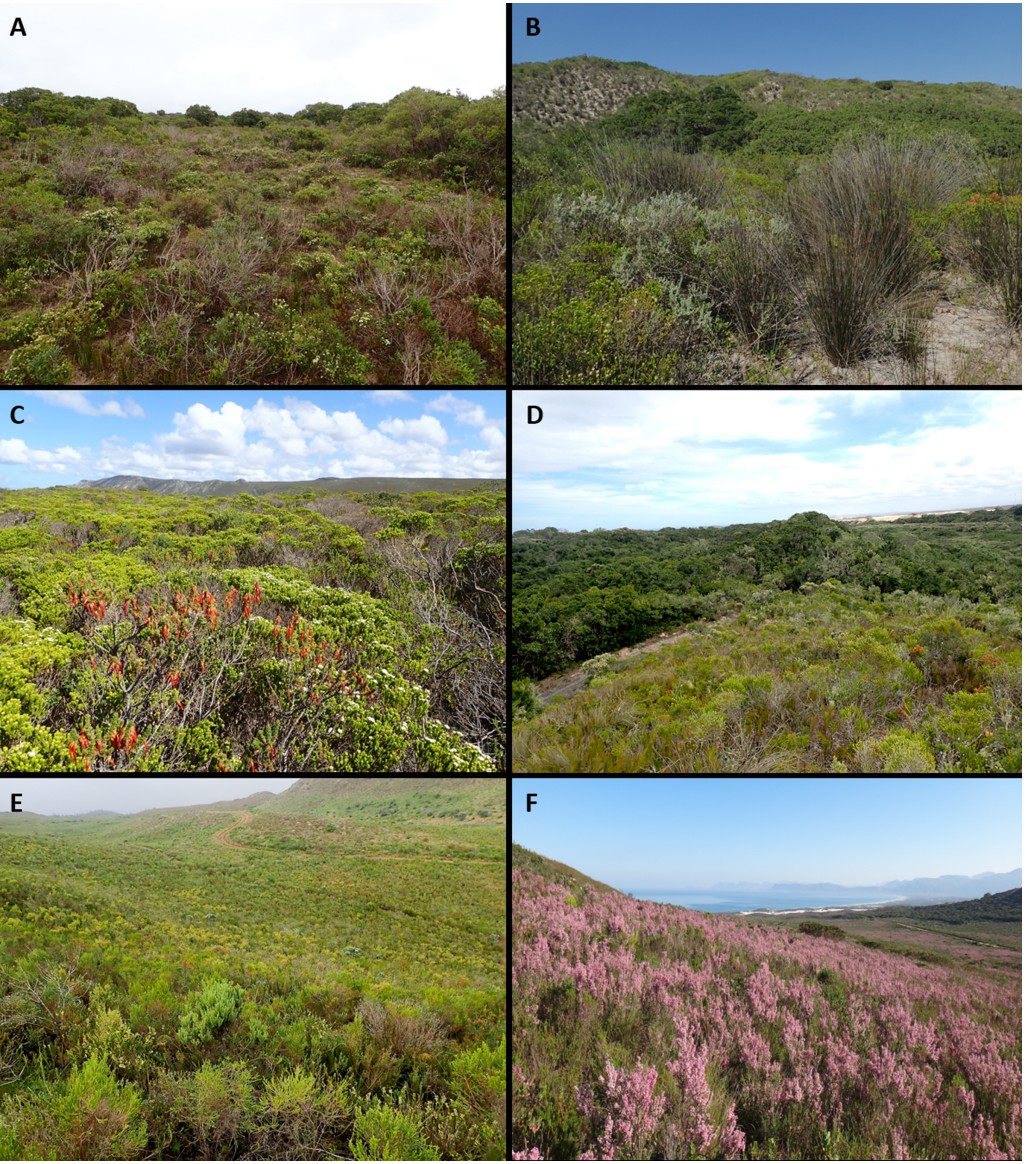

**Figure 5** **Strandveld communities of the Holocene dunes of the Cape south coast.** (A) Southeastern Strandveld at the Gamtoos River Mouth. Dominant species are *Agathosma apiculata, Restio eleocharis* and *Rapanea gilliana*. Xeric Dune Thicket occurs as bushclumps in the fynbos matrix. (B) Southwestern Strandveld east of Arniston. Dominant species are *Thamnochortus insignis, Eriocephalus racemosus* and *Agathosma collina*. Xeric Dune Thicket grows at the base of the north-facing dune ridge in the background. (C) Southwestern Strandveld west of Pearly Beach. Dominant species are *Agathosma collina, Erica coccinea* ssp. *uniflora* and *Passerina palacea*. (D) St Francis Strandveld west of Cape St Francis. Dominant species are *Passerina corymbosa, Schoenus australis, Restio leptoclados, Olea exasperata* and *Agathosma stenopetala*. Mesic Dune Thicket occupies the dune swale. (E) Goukamma Strandveld near Sedgefield. Dominant species are *Passerina corymbosa, Erica glumiflora, Metalasia muricata, Imperata cylindrica, Restio eleocharis* and *Struthiola argentea*. (F) Grootbos Strandveld near Gansbaai. Dominant species are *Erica irregularis, Indigofera brachystachya, Pentameris colorata* and *Pelargonium betulinum*. Note the large patch of Mesic Thicket in the background right, Southwestern Strandveld adjacent to mobile dunefield in the background centre and scattered individuals of *Leucadendron coniferum* close to the boundary with Limestone Fynbos upslope. Photo credits: (A–C) Adriaan Grobler; (D–E) Richard Cowling; (F) Sean Privett.

**Table 3 Spatial overlap between South Coast Strandveld units described in this study and National Vegetation Map (VEGMAP 2018) units** (*SANBI, 2018*). Percentage overlap is indicated with respect to total surface area of VEGMAP 2018 types and South Coast Strandveld types described in our study (VEGMAP 2018 types / our types).

| This study | VEGMAP 2018 biome | VEGMAP 2018 type | Overlap (%) |
|---|---|---|---|
| Goukamma strandveld | Subtropical thicket | Goukamma dune thicket | 41 / 97 |
| | Fynbos | Knysna Sand Fynbos | <1 / 1 |
| | Fynbos | Tsitsikamma Sandstone Fynbos | <1 / 1 |
| | Fynbos | Garden Route Shale Fynbos | <1 / <1 |
| | Fynbos | Southern Cape Dune Fynbos | <1 / 1 |
| Grootbos Strandveld | Fynbos | Overberg Dune Strandveld | 10 / 86 |
| | Fynbos | Agulhas Limestone Fynbos | 2 / 14 |
| | Fynbos | Overberg Sandstone Fynbos | <1 / <1 |
| Southeastern Strandveld | Subtropical Thicket | St Francis Dune Thicket | 30 / 67 |
| | Savanna | South Eastern Coastal Thornveld | <1 / 11 |
| | Fynbos | Algoa Sandstone Fynbos | 3 / 10 |
| | Albany Thicket | Hartenbos Dune Thicket | 1 / 5 |
| | Albany Thicket | Sardinia Forest Thicket | 19 / 4 |
| | Albany Thicket | Sundays Mesic Thicket | <1 / 2 |
| | Azonal Vegetation | Albany Alluvial Vegetation | <1 / <1 |
| | Fynbos | Garden Route Granite Fynbos | <1 / <1 |
| | Fynbos | Groot Brak Dune Strandveld | 1 / <1 |
| | Fynbos | Mossel Bay Shale Renosterveld | <1 / <1 |
| | Subtropical Thicket | Sundays Valley Thicket | <1 / <1 |
| Southwestern Strandveld | Fynbos | Overberg Dune Strandveld | 74 / 79 |
| | Subtropical Thicket | Hartenbos Dune Thicket | 7 / 15 |
| | Fynbos | Blombos Strandveld | 43 / 3 |
| | Fynbos | Agulhas Limestone Fynbos | 2 / 1 |
| | Fynbos | De Hoop Limestone Fynbos | 1 / 1 |
| | Subtropical Thicket | Gouritz Valley Thicket | <1 / <1 |
| | Fynbos | Albertinia Sand Fynbos | <1 / <1 |
| | Fynbos | Canca Limestone Fynbos | <1 / <1 |
| | Fynbos | Eastern Ruens Shale Renosterveld | <1 / <1 |
| | Fynbos | Groot Brak Dune Strandveld | <1 / <1 |
| | Fynbos | Potberg Sandstone Fynbos | <1 / <1 |
| St Francis Strandveld | Subtropical Thicket | St Francis Dune Thicket | 34 / 94 |
| | Fynbos | Southern Cape Dune Fynbos | 6 / 5 |
| | Fynbos | Tsitsikamma Sandstone Fynbos | <1 / 1 |
| | Subtropical Thicket | Sundays Valley Thicket | <1 / <1 |
| | Fynbos | Eastern Coastal Shale Band Vegetation | <1 / <1 |
| | Savanna | South Eastern Coastal Thornveld | <1 / <1 |

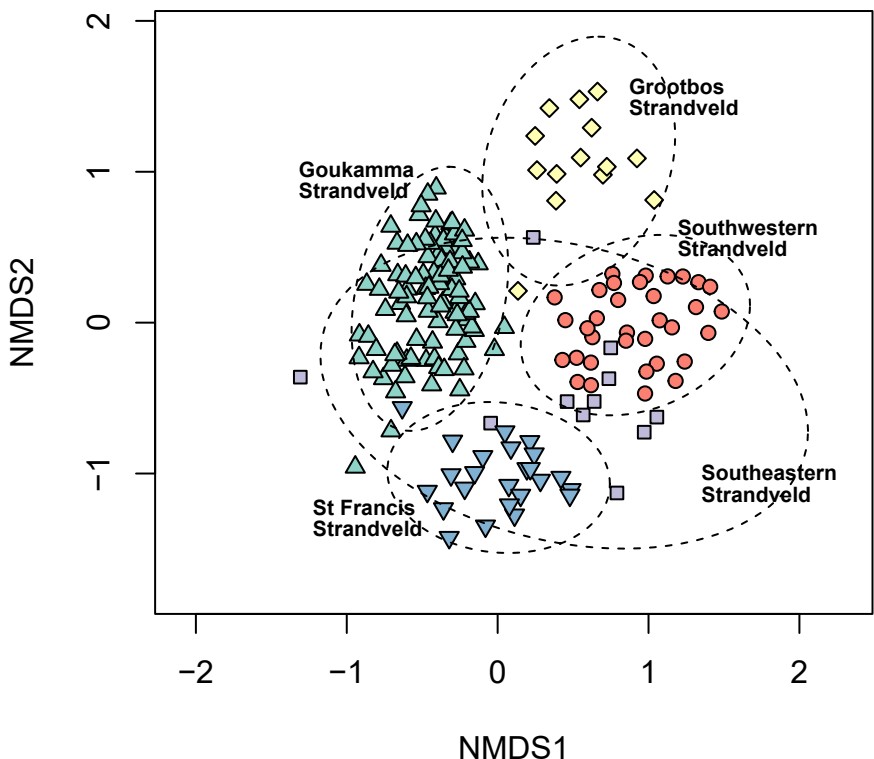

**Figure 6** Non-metric multidimensional scaling (NMDS) of the Fynbos plots sampled in this study showing their association with the five South Coast Strandveld vegetation types identified in this study. NMDS solution: $k = 2$; stress = 0.251; non-metric $R^2 = 0.937$; linear $R^2 = 0.709$.

**Table 4** Diversity and edaphic (calcareous dune sand) endemism of Dune Thicket and Strandveld vegetation units of the Cape south coast.

|  | No. plots (n) | No. species (S) | β diversity (S/n) | No. endemics (d) | % endemism (d/S*100) |
|---|---|---|---|---|---|
| **Strandveld** |  |  |  |  |  |
| Southeastern | 11 | 104 | 9.45 | 25 | 24.0 |
| St Francis | 24 | 156 | 6.50 | 29 | 18.6 |
| Goukamma | 97 | 84 | 0.86 | 14 | 16.7 |
| Southwestern | 34 | 180 | 5.29 | 27 | 15.0 |
| Grootbos | 14 | 147 | 10.50 | 14 | 9.5 |
| **Dune Thicket** |  |  |  |  |  |
| Mesic | 60 | 140 | 2.33 | 8 | 5.71 |
| Xeric | 13 | 141 | 10.85 | 26 | 18.43 |

Moreover, this unit has the highest proportion of calcareous dune endemics amongst South Coast Strandveld units, comprising 24.0% of the sample flora.

### St Francis Strandveld

St Francis Strandveld (Fig. 5D) extends from the Kromme River on St Francis Bay to the Klasies River in the eastern Tstitsikamma (Fig. 2). A smaller enclave occurs north of the Maitland River, extending towards the Gamtoos River Mouth. This strandveld unit occupies ca. 97 km². Our delimitation overlaps with the western portion of *Grobler et al.*'s (*2018*) St Francis Dune Thicket and extends marginally into *Rebelo et al.*'s (*2006*) Southern Cape Dune Fynbos between Oyster Bay and Klasies River (Tables 2, 3). Floristically, this unit is distinctive, with weak affinities to Southeastern Strandveld and Goukamma Strandveld (Fig. 6). In moist, fire-protected sites, St Francis Strandveld supports patches of Mesic Dune Thicket that approach a dune forest in both composition and stature (*Strydom et al., 2022*).

The bulk of St Francis Strandveld is associated with the broad dunefield between Oyster Bay in the west and St Francis Bay and Seal Bay in the east. The vegetated dunes are of the parabolic type with parallel dune ridges interrupted by broad to narrow troughs, some of which support wetlands. The area also supports two mobile headland bypass dunes, one with its source at Osyter Bay, the other at Thys Bay. The Holocene sediments (Schelm Hoek Formation) overlie Late Pleistocene calcarenites of the Nahoon Formation, Table Mountain Group quartzites and Bokkeveld Group shales. West of Oyster Bay, the cordon of parabolics is relatively narrow, but Holocene dunes have accumulated as wind rift dunes on the coastal margin of fairly massive and softly sculptured dunes comprising the Cenozoic sediments of the Nanaga Formation. As is the case along the eastern shores of St Francis Bay, these aeolianites support a grassy form of Sand Fynbos with a strong thicket and forest component (Fig. S1A). The enclave of St Francis Strandveld at Maitland is also partly perched on Nanaga sediments.

The climate of St Francis Strandveld is similar to Southeastern Strandveld with the exception of considerably higher annual rainfall experienced west of Seal Point at Cape St Francis (Fig. 1). Here rainfall ranges from ca 700 mmyr$^{-1}$ in the east to almost 1,000 mmyr$^{-1}$ in the west, the wettest conditions experienced by Strandveld in the CFR.

Locally dominant species differential for St Francis Strandveld include the shrubs *Agathosma stenopetala* and *Phylica litoralis,* and the restioid *Restio leptoclados* (Fig. 4, Table S1). Commonly encountered differential species of low dominance comprise the shrublets *Indigofera stricta* and *Tephrosia capensis* and the geophytes *Satyrium princeps* and *Othonna rufibarbis* as well as numerous less frequent species, including shrubs *Erica chloroloma* (also encountered albeit rarely in Southeastern Strandveld), *Cullumia decurrens* and *Indigofera sulcata*, and forbs *Manulea obovata* and *Hebenstreitia integrifolia*. Mesic Dune Thicket patches are widespread throughout St Francis Strandveld where they are associated with moist dune troughs. Dominant species are *Pterocelastrus tricuspidatus, Mystroxylon aethiopicum* and *Sideroxylon inerme* (*Cowling, 1984*; *Strydom et al., 2022*).

We recorded 156 species in 24 plots, producing a moderate $\beta$-diversity score of 6.50 (Table 4). This sample yielded 29 species endemic to calcareous coastal dunes, 18.6% of the total.

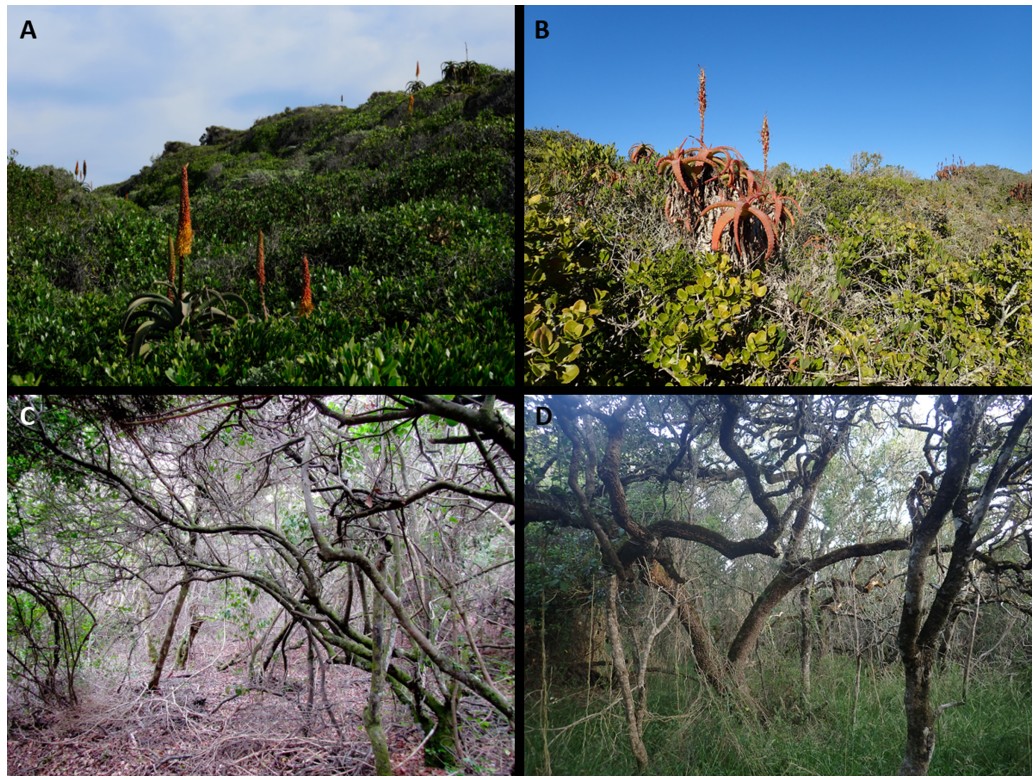

**Figure 7** **Thicket communities of the Holocene dunes of the Cape south coast.** (A) Xeric Dune Thicket at Cape Recife south of Gqeberha. Dominant species are *Sideroxylon inerme* and *Aloe africana*. (B) Xeric Dune Thicket near Mossel Bay. Dominant species are *Mystroxylon aethiopicum, Aloe arborescens, Schotia afra* and *Rhoicissus digitata*. (C) Mesic Dune Thicket near Sedgefield. Dominant species are *Cassine peragua, Sideroxylon inerme, Cussonia thyrsiflora* (liana) and *Pterocelastrus tricuspidatus*. (D) Mesic Dune Thicket near Gansbaai. Dominant canopy species are *Sideroxylon inerme* (multi-stemmed) and *Chionanthus foveolatus* (single stemmed). *Stipa dregeana* is prominent in the field layer. Photo credits: (A, B) Adriaan Grobler; (C, D) Richard Cowling.

### *Goukamma Strandveld*

Goukamma Strandveld (Fig. 5E) is associated with the Holocene dunes (Strandveld Formation) of the Wilderness-Sedgefield embayment between the Knysna Heads in the east and Wilderness in the west (Fig. 2) where it covers ca. 39 km$^2$. Minor and floristically depauperate enclaves occur at Nature's Valley and Plettenberg Bay. Floristically, the unit shows weak floristic overlap with St Francis Strandveld and Grootbos Strandveld (Fig. 6). Like this unit, it supports numerous patches of Mesic Dune Thicket that attains forest stature in fire-protected and locally moist sites (Fig. 7E).

Parabolic dunes occur along the coastal margin, for example between Buffalo Bay and the Goukamma River, where they overlie calcarenites of the Waenhuiskrans Formation (cf *Bateman et al., 2011*). Elsewhere the sediments have accumulated as wind rift dunes on the seaward ridge of the multi-dune Wilderness-Sedgefield embayment. Here they also mantle sediments of the Waenhuiskrans Formation, which comprise the inland ridges and support Knysna Sand Fynbos (Fig. S1B).

The climate of Goukamma Strandveld is like St Francis Strandveld but with lower annual rainfall (500–700 mmyr$^{-1}$) (Fig. 1) and a considerably more subdued wind regime.

Differential dominant shrub species for Goukamma Strandveld are *Cliffortia falcata*, *Chironia decumbens* and *Erica glandulosa* ssp. *fourcadii* (Fig. 5E, Table S1). Other frequently encountered differential shrubs include *Disparago kraussii*, *Carpobrotus edulis*, *Cliffortia linearifolia* and *Lachnaea diosmoides*. While these species distinguish Goukamma amongst Strandveld units, all of them are common in adjacent Knysna Sand Fynbos. Mesic Dune Thicket patches are widespread in Goukamma Strandveld (Fig. 7E).

Goukamma Strandveld has a relatively depauperate flora; we recorded only 84 species in 94 plots, yielding a $\beta$-diversity of 0.86 (Table 3), the lowest of any of the vegetation units in this study. This may be due to the small extent of this unit. However, this pattern is consistent with the low diversity recorded at a variety of spatial scales in fynbos in the so-called forest gap located between George and the eastern Tstitsikamma (*Weimarck, 1941*; *Cowling, 1983*). Our sample included only 14 calcareous dune endemics, comprising 16.7% of the flora.

### Southwestern Strandveld

Southwestern Strandveld (Figs. 5B and 5C) is associated with the Holocene dune cordon between Mossel Bay and Cape Hangklip, except for the sheet of calcareous sand northeast of Die Kelders, which supports Grootbos Strandveld (Fig. 2). As such, it includes part of the coastal margin of *Grobler et al.*'s (*2018*) Hartenbos Dune Thicket, the western sector (between the Duiwenhoks and Breede Rivers) of *Rebelo et al.*'s (*2006*) Blombos Strandveld and, except for the Grootbos Strandveld enclave, all the Overberg Dune Strandveld (Tables 2, 3). This is the most expansive of our strandveld units, occupying ca. 322 km$^2$. Southwestern Strandveld shows strongest florisitc links to Grootbos Strandveld and Southeastern Strandveld (Fig. 6). The extensive thicket patches in this unit are mostly Xeric Dune Thicket although small patches of *Sideroxylon inerme*-dominated Mesic Dune Thicket do occur locally in favourable sites, especially west of Cape Agulhas (*Boucher, 1978*).

Like Southeastern Strandveld, much of this unit occurs on a narrow dune cordon. However, there are extensive dune fields at De Hoop, Cape Agulhas and Walker Bay, including large areas of mobile sand, much of which has been artificially stabilised. The vegetated dunes are dominated by parabolic forms. However, hummock-blowouts and playa lunette dunes of Holocene age (*Carr, Thomas & Bateman, 2006*) are uniquely found in the high wind energy environment west of Cape Agulhas (*Tinley, 1985*). The Holocene dunes (Strandveld Formation) rest on Early Pleistocene calcarenites and neutral sands (Waenhuiskrans Formation) and Table Mountain Group sandstones. Sand Fynbos and Limestone Fynbos abut the inland margin of Southwestern Strandveld, associated with Waenhuiskrans unconsolidated and cemented sediments, respectively (Figs. S1C–S1D).

Southwestern Strandveld encompasses the transition from a non-seasonal to a weakly winter rainfall regime. Between Mossel Bay and Arniston (Fig. 2), rainfall is highest and most reliable in the equinoctial months, but with an increasing winter component westward. Furthermore, rainfall totals are at their lowest along the Cape south coast,

notably between Mossel Bay and Vlees Bay, and between De Hoop and Arniston, where totals barely reach 400 mmyr$^{-1}$. West of Arniston, most rain falls in the winter months and totals range between 450 and 500 mmyr$^{-1}$. Here summer temperatures are slightly warmer and winter temperatures slightly cooler than elsewhere on the Cape south coast (*Tinley, 1985*). The wind regime is fierce, especially from Cape Agulhas to Danger Point where winter westerlies have a strong northerly component.

Southwestern Strandveld has a rich array of differential species, especially if Cape species associated with the patches of embedded Xeric Dune Thicket are included. Dominant differential shrub species include *Helichrysum retortum, Passerina paleacea, Agathosma collina, A. dielsiana, A. geniculata, Acmadenia obtusata, Disparago anomala, Lampranthus fergusoniae, Indigofera calciphila, I. harveyi, Erica bredasiana, E. vernicosa, E. coccinea* ssp. *uniflora, Argyrolobium harmsianum, Ruschia macowanii* and *Berkheya coriacea* (Fig. 4, Table S1). Xeric Dune Thicket patches in Southwestern Thicket have a high cover of *Roepera morgsana, Euphorbia burmannii* and *E. mauritanica*, showing a floristic link to the Strandveld thicket of the Cape west coast, where these species are abundant (*Boucher & Jarman, 1977*).

The Southwestern Strandveld flora is rich: we recorded 180 species in 34 plots, producing a $\beta$-diversity score of 5.29 (Table 4). This is to be expected, given that this unit spans a long section of the Cape coast, including the transition from non-seasonal to winter rainfall. Of our sample flora, 15.0% are calcareous coastal dune endemics.

### Grootbos Strandveld

Grootbos Strandveld (Fig. 5F) occupies ca. 40 km$^2$ between Die Kelders and Stanford on the shores of Walker Bay (*Mergili & Privett, 2008*), making it, after Goukamma Strandveld, the second smallest Strandveld unit recognised in this study. It is associated with a sheet of Holocene sediment inland of the Walker Bay mobile dunefield. We suspect these sediments are older than those supporting Southwestern Strandveld, something that can be resolved with OSL dates. These calcareous sands, which overlie neutral, yellow-brown sands of the Waenhuiskrans Formation, are thin on their inland margin, thus blurring the distinction between *Leucadendron coniferum*-dominated Sand Fynbos (Fig. S1D) and Grootbos Strandveld (*Mergili & Privett, 2008*). Where these sands abut hills of Table Mountain Group sandstone and are underlain by Waenhuiskrans calcarenites that impede water drainage, soil moisture conditions are favourable for the development of Mesic Dune Thicket (Fig. 7D).

The climate of Grootbos is like that of the western sector of Southwestern Strandveld. The predominantly winter rainfall amounts to about 500 mmyr$^{-1}$.

Although Grootbos Strandveld shows strong floristic overlap with Southwestern Strandveld (Fig. 5), it nonetheless has a distinctive flora including many differential species (Fig. 4, Table S1). Notable amongst these is *Erica irregularis,* a dominant species that is entirely confined to this unit. Other dominant differential shrub species include *Indigofera brachystachya, Diosma subulata, Olea capensis* (a distinctive form largely endemic to the Agulhas Plain), *Clutia polygonoides* and *Agathosma imbricata*. Differential graminoids include *Thamnochortus erectus, Ficinia indica* and *Pentameris colorata*. On its inland margin
the spill over from the adjacent Sand Fynbos results in populations of *Leucadendron coniferum, Leucospermum patersonii* and other species not normally found on calcareous sands (*Mergili & Privett, 2008*).

Grootbos Strandveld is the richest among all Strandveld units, having 147 species in 14 plots and the highest $\beta$-diversity score (10.50) (Table 4). However, at 9.52%, the proportion of calcareous dune sand endemics is the lowest. Both patterns could be a consequence of the large numbers of Sand Fynbos species that are found on the inland fringes of Grootbos Strandveld.

## Subtropical Thicket Biome vegetation

Throughout its range, South Coast Strandveld includes patches of unbroken thicket of variable size, which are floristically well differentiated from the fynbos matrix in which they are embedded (Fig. 3). While some of these thicket patches are sufficiently large to be mapped at the VEGMAP scale (1:000 000) (*Mucina et al., 2006a*; *Mucina et al., 2006b*), we include these units as Fynbos Biome vegetation within the corresponding Strandveld unit that is defined based on its corresponding non-thicket (fynbos) flora.

The size and location of these patches depends on soil moisture and exposure to fire, thicket generally preferring wetter sites with low exposure to fire, for example steep-sided (fire-protected) dune hollows and the littoral margin where groundwater approaches the surface (*Cowling, 1984*; *Strydom et al., 2022*). Except for some vertically growing species more typical of true forest, dune thicket species tolerate fire *via* vigorous resprouting (*Strydom et al., 2020*; *Strydom et al., 2021*). Evidence suggests that thicket patches are relatively stable over decades-long intervals (*Cowling & Hoffman, 2021*; *Strydom et al., 2022*). However, this is not to say that frequent fire coupled with browsing by megaherbivores would not have limited the extent of thicket in suitable habitats in the dune landscape (*Strydom et al., 2023*).

Dune thicket of the Cape south coast is distinguished from other Subtropical Thicket Biome vegetation by the consistently high cover of *Sideroxylon inerme, Rhoicissus digitata, Mystroxylon aethiopicum* and *Chionanthus foveolatus* (Fig. 4, Table S1) (*Vlok, Euston-Brown & Cowling, 2003*; *Mucina et al., 2006a*; *Mucina et al., 2006b*). *Pterocelatrus tricuspidatus*, a common tree of south Cape non-dune thicket and Afromontane forest, is also invariably present as a multi-stemmed shrub. Dune thicket is further differentiated by a suite of species largely endemic to calcareous coastal substrata. These include *Olea exasperata, Robsonodendron maritimum, Euclea racemosa, Maytenus procumbens* and *Rapanea gilliana* (east of Tsitsikamma River) (*Cowling, 1983*). Generally, the diversity of dune thicket species declines from east to west within the CFR (*Tinley, 1985*).

We recognised two dune thicket communities in the study region, namely Mesic Dune Thicket and Xeric Dune Thicket (Fig. 8, Table S1).

### *Mesic Dune Thicket*

As the term applies, Mesic Dune Thicket (Figs. 7C–7D) occupies sites with access to high levels of soil moisture. It occurs throughout the study region but is poorly represented in the Southeastern Strandveld and Southwestern Strandveld. We recognised three subunits,

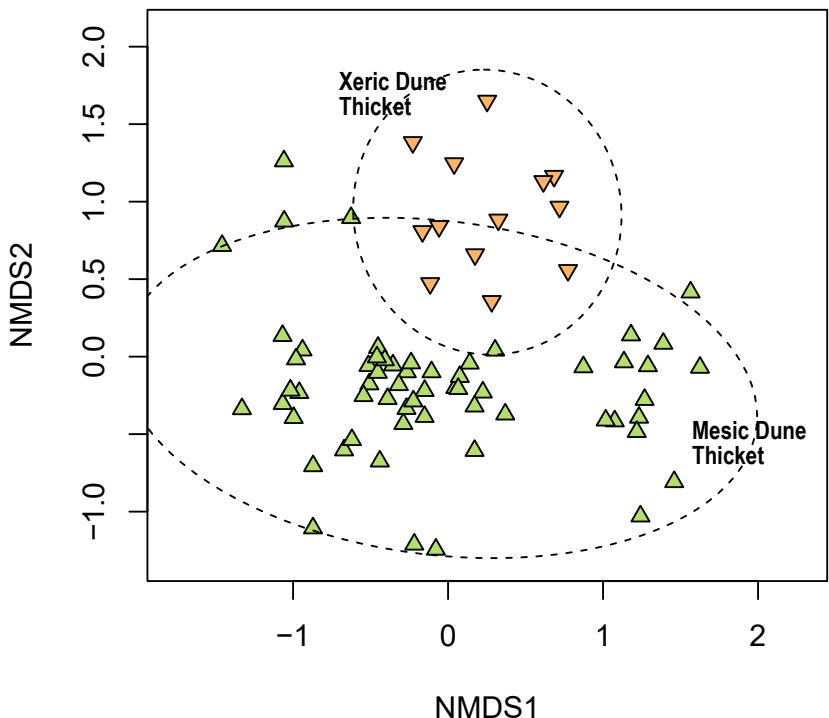

**Figure 8** Non-metric multidimensional scaling (NMDS) of the Thicket plots sampled in this study showing their association with the two dune thicket vegetation types identified in this study. NMDS solution: $k = 2$; stress = 0.199; non-metric $R^2$ = 0.960; linear $R^2$ = 0.819.

two eastern ones (at Cape St Francis and Goukamma), and a western one (at Grootbos) (Fig. S2). While dominant species have a multi-stemmed, laterally spreading architecture (*e.g.*, *Sideroxylon inerme*, *Pterocelastrus tricuspidatus*), single-stemmed, vertically-growing species are indicative, for example *Zanthoxylum capense*, *Apodytes dimidiata*, *Celtis africana*, *Clausena anisata*, *Afrocanthium mundianum* and *Acokanthera oppositifolia* (*Strydom et al., 2022*) (Fig. 4, Table S1). Canopy height is in the order of 4–6 m. Mesic Dune Thicket usually has a well-developed herbaceous understorey comprising *inter alia* *Brachiaria chusqueoides*, *Hypoestes aristata*, *Amaranthus thunbergii*, *Droguetia iners* and *Stipa dregeana*. The liana and vine floras are rich with the most common and widespread species being *Asparagus scandens*, *Capparis sepiaria*, *Dioscorea mundii*, *Secamone alpini*, *Behnia reticulata* and *Kedrostis nana*.

Mesic Dune Thicket has a relatively impoverished flora; we recorded 140 species in 60 plots, yielding a $\beta$-diversity score of 2.33, the lowest amongst all vegetation units described here (Table 4). Edaphic endemism was also the lowest, at 5.71%. These patterns are likely a consequence of the tropical affinity of dune thicket lineages and their increasing depauperization westwards into the cool-season rainfall regime of the western CFR (*Cowling, 1983*; *Tinley, 1985*).

### Xeric Dune Thicket

Xeric Dune Thicket (Figs. 7A–7B), which occurs throughout the study region, is the dominant dune thicket in the drier part of the study region along the eastern shores of St Francis Bay and between Groot Brak River and Cape Agulhas. Elsewhere, it is confined to edaphically dry sites, *e.g.*, north-facing dune slopes. We were unable to recognise any subunits within Xeric Dune Thicket (Fig. S2), suggesting considerable floristic homogeneity of dominant species along a 700-km stretch of coast. Multi-stemmed, hedge-forming shrubs are the dominant architecture, although lateral spreaders may be locally common (*Strydom et al., 2021*; *Strydom et al., 2022*). The Xeric Dune Thicket canopy is extremely dense and intercepts most light and rainfall; this may explain the absence of a well-developed understorey. Canopy height varies from 0.5–3 m. Dominant canopy species comprise a diverse assemblage including *Pterocelastrus tricuspidatus, Sideroxylon inerme, Euclea racemosa, Searsia glauca, S. crenata, S. pterota, Colpoon compressum, Maytenus procumbens, Olea exasperata* and *Putterlickia pyracantha* (Table S1). Species that differentiate the xeric from mesic forms of dune thicket include *Robsonodendron maritimum, Searsia pterota, Zygophyllum morgsana, Lycium ferocissimum, Azima tetracantha, Euphorbia mauritanica* and *E. burmannii* (Fig. 4, Table S1). Xeric Dune Thicket shows strong floristic similarities to the thicket component of Strandveld along the Cape west coast (cf. *Boucher & Jarman, 1977*).

Owing to its low and fragmented structure that enables the admixture of fynbos and thicket lineages in Xeric Dune Thicket, as well as its extent throughout the study region (Cape Recife to Cape Hangklip), this unit is relatively species-rich (141 species in 13 plots) and also having both high $\beta$-diversity (10.85) and high edaphic endemism (18.43%) (Table 4).

## Vegetation–sediment-age relationships

Based on soil features, dune morphology and vegetation, what we term Holocene dunes represent a coherent geo-botanical unit. As we pointed out above, geology maps classify much of what we identify as Holocene dunes as Late Pleistocene or even Neogene sediments. OSL-dated sediments are rare in our study area but do confirm Holocene age for the sediments at the sites we sampled at Goukamma (*Ben Arous, Duval & Bateman, 2022*; *Bateman et al., 2004*; *Bateman et al., 2011*) and on the Agulhas Plain (*Carr, Thomas & Bateman, 2006*). We suspect that the fixed, calcareous dunes west of Cape St Francis and at Grootbos are of early Holocene to terminal Pleistocene age, but this hypothesis remains to be tested.

Problems with the assignation of sediments to deposition sequences may arise because of the stacking of serial dune depositions as has been described in the Wilderness-Sedgefield embayment by *Bateman et al. (2004)* and *Bateman et al. (2011)* but also evident at Cape St Francis (*Tinley, 1985*) and elsewhere (*e.g.*, Walker Bay). Globally, sea-level fluctuations of the Pliocene and Pleistocene have shifted palaeoshorelines between elevations of over 10 m above present, to ~130 m below present base level (*Spratt & Lisiecki, 2016*). South Africa's tectonic position and resultant geological record during this period generally mirrors these glacio-eustatic trends (*Cawthra et al., 2018*). With the highstand deposits preserved along
the modern coast, and sampling and observational bias allowing us to work on these and less so under the sea, neo-coastal deposits show that it is apparent that shorelines have repeatedly returned to the same elevations. When a stable shoreline ensues and sediment is supplied to that coast, it tends to either prograde outward in a seaward direction, or accumulate on top of existing deposits where accommodation space is limited. In the Cape, the latter is often the case as accommodation of sediment is hindered by steep bedrock cliffs and mountains in the immediate hinterland. As such, stacking can occur when there is no room for adjacent packing of younger sediments. Thus, Holocene deposits may overlie older aeolianites but this does not mean that they have weathered from these deposits.

However, as already pointed out above, Holocene sands are physically and chemically different from the Pleistocene sands in many ways, most of which can be identified in the field. The vegetation of the Cape coast is highly responsive to these differences, with alkaline Holocene sand supporting a floristically distinct Strandveld vegetation with a different structure and sharing few species with the Sand Fynbos and Limestone Fynbos of the older sediments (*Cowling, 1990*). In the absence of OSL dates, geology could learn much from soil science and plant ecology when mapping Cenozoic sediment along the Cape coast.

What is clear is that coastal dune landscapes are dynamic, accommodating in some places successive phases of accretion throughout the Quaternary. These processes would juxtapose floras associated with different-aged sediments, producing novel plant communities and genotypes because of invasion onto newly available edaphic surfaces. Importantly, these communities, along with their high incidence of endemics, have been assembled over the past 6 ka. There may well be correlates between Strandveld community structure and sediment age that could be explored. However, this will have to await comprehensive dating of Holocene and Late Pleistocene coastal sediments of the Cape south coast.

## Role of fire exposure and soil moisture in the coexistence of fynbos and thicket cover states

As we have shown above, South Coast Strandveld comprises a mosaic of open-canopy fynbos and closed-canopy thicket, each comprising a floristically and phylogenetically distinct assemblage. The relative abundance of these two formations is a function largely of fire and moisture regimes, and their interactions; however, browsing by megaherbivores in the precolonial era likely also played an important but undocumented role.

Under high moisture regimes, for example dune troughs where the water table is close to the surface, or high annual rainfall regimes (>ca. 750 mm), and where fire is excluded or occurs at century-long intervals, the Holocene dune landscape would likely be dominated by Mesic Dune Thicket and Dune Forest (*Strydom et al., 2022*) (Fig. 9). The corresponding dominant architectural types of canopy species are lateral spreaders (*e.g.*, *Sideroxylon inerme*) and vertical growers (*e.g.*, *Chionanthus foveolatus*) (*Strydom et al., 2020*). At the other extreme, where fire frequency is high (ca. 20-year intervals) and the moisture regime is low (annual rainfall <ca. 450 mm), the dune landscape is dominated by fynbos with patches of Xeric Dune Thicket comprising low, multi-stemmed (hedge-forming), geoxylic (with underground stems) shrubs (*e.g.*, *Robsonodendron maritimum, Olea exasperata*) that

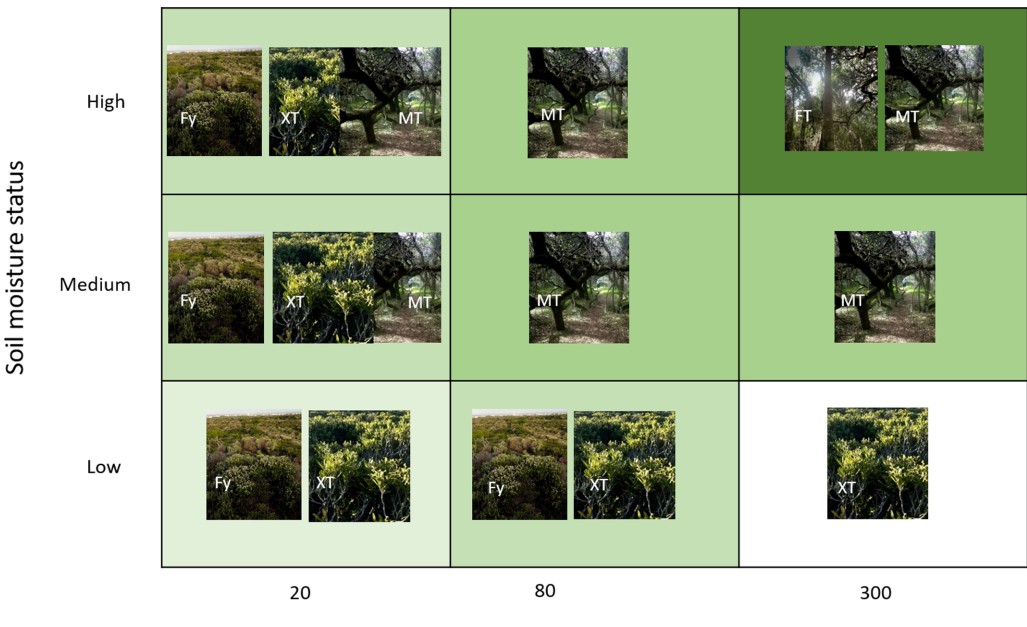

**Figure 9 Hypothetical depiction of vegetation of Holocene dunes in relation to categories of soil moisture status and fire exposure.** More than one unit per category combination is indicative of a mosaic structure. Fy = Fynbos, FT = Forest-Thicket, MT = Mesic Thicket, XT = Xeric Thicket. Photo credits: Richard Cowling.

are restricted to relatively moist sites such as high-water table dune troughs and steep, south-facing dune slopes. Under a high-moisture, high-fire frequency combination, hedge thicket is replaced in the most favourable sites by a taller thicket dominated by lateral spreaders such as *Pterocelatrus tricuspidatus* and *Mystroxylon aethiopicum* (*Strydom et al., 2022*). The extent to which this occurs will depend on the frequency of fires with sufficient intensity to consume these taller thicket patches. Such fires, by reducing canopy height and promoting vigorous sprouting of thicket species from below-ground branches, are key to maintaining the relative mix of thicket dominated by hedge formers and lateral spreaders (*Strydom et al., 2021*; *Strydom et al., 2022*). Our model predicts that in fire-free dune landscapes of uniformly low soil moisture, hedge thicket would dominate.

Fynbos is predicted to be a stable cover state on Holocene dunes where high-intensity fires occur at ca. 20-year intervals, irrespective of soil moisture regime; only on sites of low soil moisture does fynbos persist at ca. 80-year intervals but is outcompeted by Xeric Thicket where fires occur at centuries-long intervals (Fig. 9). This is consistent with the requirement for fire-stimulated recruitment of the generally short-lived fynbos lineages both in the Holocene dune landscape (*Kilian & Cowling, 1992*; *Pierce & Cowling, 1991a*; *Pierce & Cowling, 1991b*; *Pierce & Moll, 1994*) and elsewhere in the CFR (*Kraaij & Van Wilgen, 2014*).

## Conservation implications

Coastal dunes are particularly fragile ecosystems, as even minor disturbances can cause long-term alterations and compromise biodiversity (*Martínez, Maun & Psuty, 2008*). Though specific impacts depend on local conditions (*Cazenave & Le Cozannet, 2014*), rising sea levels driven by climate change are likely to result in the inland migration of the beach–dune profile (*Feagin, Sherman & Grant, 2005*; *Pethick, 2001*; *Psuty & Silveira, 2010*). Dune ecosystems may remain intact and functional even after such a displacement, but this is entirely dependent on the existence of sufficient accommodation space on the landward margin (*Psuty & Silveira, 2010*). Should this not be the case, a "coastal squeeze" will ensue, leading to the extirpation of dune habitats and their component species (*Feagin, Sherman & Grant, 2005*; *Martínez et al., 2014*; *Mendoza-González et al., 2013*). The risk of this is especially high along developed coasts, where hard infrastructure effectively limits the extent to which coastal ecosystems can migrate inland in response to sea-level rise. Ecological niche modelling, considering predicted climate change and sea-level rise, suggests that the distribution of coastal plant species in Mexico will become severely limited, and especially so for local endemics (*Mendoza-González et al., 2013*). Similarly, fixed-dune habitats in Italy, being more sensitive to direct climate change effects, are projected to lose most of their distribution in the coming decades (*Prisco, Carboni & Acosta, 2013*).

While much more research is needed to understand the potential impacts of climate change and sea-level rise on coastal dune vegetation on the southern coast of the CFR, we expect similar ramifications to those predicted for dunes elsewhere. All South Coast Strandveld units support dune endemics with small ranges (*Grobler & Cowling, 2021*), many of which are already threatened by coastal development and invasive alien plants, especially *Acacia cyclops* (*Cowling et al., 2019*); given projected global change, these localised edaphic endemics are most likely to suffer further range contractions (*Xu et al., 2023*). Adding to this, ramifications of coastal squeeze are likely to be exacerbated by the fact that inland habitats (*e.g.*, sand fynbos) are unlikely to be a reliable source of propagules for impacted strandveld vegetation as these non-dune habitats largely lack calcicolous species, which constitute about 40% of dune floras in the Cape (*Cowling et al., 2019*; *Grobler & Cowling, 2021*).

Nearly half of South Africa's coastal habitat has been degraded, while 60% of coastal ecosystem types (or 55% by area) are considered threatened (*Harris et al., 2022*). Impacts have been especially severe along the dune-supporting sandy coasts of the country, including along the Cape south coast (*Tinley, 1985*), yet the conservation importance of these coastal dune habitats has historically been underappreciated (*Cowling, 1980*; *Cowling & Pierce, 1985*). *Rouget et al. (2006)* classified South African vegetation types according to their ecosystem status, a measure based on the extent of remaining untransformed area of a vegetation type in relation to its biodiversity target (% area). In this classification, all vegetation types occurring on southern Cape coastal dunes were listed as "Least Threatened". An updated status assessment, based on the latest classification of South Africa's vegetation (*Dayaram et al., 2019*) and implementing the IUCN Red List of Ecosystems V. 1.1 protocol (*Keith et al., 2013*), classified most Cape south coast dune vegetation as "Least Concern", although Overberg Dune Strandveld and Hartenbos Dune

Thicket were classified as "Endangered" and Grootbrak Dune Strandveld as "Critically Endangered" (*Skowno & Monyeki, 2021*).

As we have explained, however, the delimitation of vegetation units on coastal dunes of the Cape south coast has hitherto been problematic, which has introduced inherent errors in the threat status assessments of these ecosystems. There is thus an urgent need for the reassessment of transformation and protection levels and threat statuses of these ecosystems to identify regional conservation priorities and to implement effective measures in these areas to prevent further degradation and loss of dune vegetation. We propose that our classification of South Coast Strandveld be used for this purpose as it allows for a more accurate understanding of the distribution and composition of coastal dune vegetation units, thereby providing a more reliable foundation for spatially explicit conservation assessments. Such assessments should further consider the regional and global rarity of these coastal dune vegetation units (*Van Der Maarel & Van Der Maarel-Versluys, 1996*): relative to the more extensive ecosystems of the southern Cape hinterland (Fig. 2), they occur as small (mostly <10 km$^2$; Fig. S3) and fragmented patches in a dynamic landscape. Thus, given the continuing threat of coastal development and encroachment by invasive plants, as well as the exacerbating effects of climate change and associated sea-level rise, we propose that all remnant South Coast Strandveld vegetation be protected to facilitate the persistence of these habitats and their component species.

## CONCLUSIONS

The vegetation of the Holocene dunes of the Cape south coast form a coherent geo-botanical unit (South Coast Strandveld) that can be further divided into six floristically differentiated units comprising mosaics of fynbos and thicket. These Strandveld units, despite their small extent, are rich in species including numerous dune endemics. They are floristically distinct from the vegetation on older, mostly Late Pleistocene sands, with which they have been conflated in previous studies. We recommend that our study be replicated on these older sediments, thereby enabling a comprehensive floristic assessment of Cenozoic sediments in the study area. We also recommend that this study be extended to the Cenozoic sediments of the Cape west coast and further afield in Namaqualand. This research should include comprehensive data on sediment ages to test the hypothesis that botanical composition of vegetation can predict accurately sediment age. Botanical transects across the full spectrum of Cenozoic coastal sediments in the Cape offer great potential for understanding the evolutionary underpinnings of Cape coastal clades, representing the youngest radiation in the hyperdiverse Cape flora (*Hoffmann, Verboom & Cotterill, 2015*).

## ACKNOWLEDGEMENTS

Our gratitude goes to Alastair J. Potts for technical assistance during early stages of the work. Thank you to Sinenjongo Gcina and Susan Botha for assistance with fieldwork.

### Funding

Funding for this research was provided by the National Research Foundation of South Africa (NRF) through the Foundational Biodiversity Information Programme (Grant No. 110438). B. Adriaan Grobler was supported by an NRF postdoctoral fellowship (Grant No. 116756) and an African Centre for Coastal Palaeoscience postdoctoral fellowship during the completion of this work. The funders had no role in study design, data collection and analysis, decision to publish, or preparation of the manuscript.

### Grant Disclosures

The following grant information was disclosed by the authors:
The National Research Foundation of South Africa (NRF) through the Foundational Biodiversity Information Programme: 110438.
An NRF postdoctoral fellowship and an African Centre for Coastal Palaeoscience postdoctoral fellowship: 116756.

### Competing Interests

Richard M. Cowling is an Academic Editor for PeerJ.

### Author Contributions

- Richard M. Cowling conceived and designed the experiments, performed the experiments, analyzed the data, prepared figures and/or tables, authored or reviewed drafts of the article, and approved the final draft.
- Hayley Cawthra analyzed the data, prepared figures and/or tables, authored or reviewed drafts of the article, and approved the final draft.
- Sean Privett performed the experiments, authored or reviewed drafts of the article, and approved the final draft.
- B Adriaan Grobler conceived and designed the experiments, performed the experiments, analyzed the data, prepared figures and/or tables, authored or reviewed drafts of the article, and approved the final draft.

### Field Study Permissions

The following information was supplied relating to field study approvals (i.e., approving body and any reference numbers):
   Denel Overberg Test Range (no permit required).

### Data Availability

   The data used in phytosociological and multivariate analyses are available in the Supplementary File.

### Supplemental Information

Supplemental information for this article can be found online at http://dx.doi.org/10.7717/peerj.16427#supplemental-information.

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
