# Peer review of "The vegetation of Holocene coastal dunes of the Cape south coast, South Africa"

_PeerJ, doi:10.7717/peerj.16427_

## Round 0.1 · original submission · Minor Revisions

We have received two thorough reviews, and even though one of the reviewers suggest major reviewer, I agree with reviewer 2 that this is a really well written article that requires minor revision.

Reviewer 1 ·

Basic reporting

The article is written in a professional English, however, some terms are not clear enough (see comments below)

Relevant literature is appropriately referenced. To improve the context provided, further references are mentioned (see comments below)

Article structure is adequate although it is excessively lengthy, Comments on the article structure are provided below.

Experimental design

This is an original research
Research questions are relevant and meaningful, although not well defined (see comments below)
This is a rigorous investigation
Methos are described with sufficent detail

Validity of the findings

Vegetation data are robust but see comments below

Additional comments

Introduction
The Introduction section is currently too brief and lacks sufficient focus on the vegetation aspect of your study, which is the central theme. It would be beneficial to expand on the relevance of vegetation in coastal dunes and its significance in the broader context of your research.
Lines 80-83. Plant oceanic dispersal is probably little studied yet, however, many species seem to have oceanic dispersal (thalassochory) so it’s probably an important mechanisms in coastal dunes. See for example, Cuena-Lombraña et al. 2022. Understanding long-distance seed dispersal by sea currents… and Miryeganeh et al. 2014. Long-Distance Dispersal by Sea-Drifted Seeds Has Maintained the Global Distribution of Ipomoea pes caprae...
Certain terms used in the manuscript need further clarification. For instance, in Lines 89-91, the phrase "defensible treatment of the dune" and "habitat retention" require more explicit explanations to aid readers in understanding their relevance. Furthermore, the term "intact" when describing habitats should be clarified as may not represent the prevailing conditions in most cases.
Additionally, the scopes of the paper lack full specification, probably you should explicitly mention vegetation mapping as part of the study.

Materials and Methods:
The section on Materials and Methods provides an adequate description of your approach, particularly your utilization of the phytosociological approach. However, Lines 301-313 can be omitted or significantly condensed as the advantages of this approach are widely known among vegetation scientists. Instead, I suggest moving the paragraph (Lines 315-319) that outlines your unique methodology to the Introduction section. This relocation would emphasize the significance of your study from the outset.

Results and Discussion
The presentation of results and discussion requires improvement for better clarity and readability. The first page of this section is advocated to discuss the Strandveld concept (Acocks 1953), a term embedded in the Cape Afrikaans vernacular dating back to the Dutch settlement of the Cape coasts. This part could be synthetized.
It is essential to organize this section in a more structured manner, starting with the results of data analyses, such as the NMDS ordination. Clearly describe the outcomes of these analyses before delving into the discussion and interpretation of the results.
The last two sections, i) vegetation–Sediment-Age Relationships and ii) Role of Fire Exposure and Soil Moisture in the Coexistence of Fynbos and Thicket Cover States are essentially interpretations based on literature, so they could be also synthetized.

Conservation Implications and Conclusions.
No further comments on these sections.

·

Basic reporting

All excellent.

Thorough and extensive background and referencing of both substrates and vegetation description history is in itself a valuable resource much appreciated by this reviewer.

Minor stylistic changes suggested but none critical other than the typos - see attached doc.

Experimental design

Substantial volume of field survey data of high quality appropriate for the scope and conclusions. Solid appropriate analysis using accepted approaches.

Validity of the findings

Solid and well documented. All data except GIS electronic mapping data provided.

Additional comments

See attached doc
Overall – important and valuable foundational research paper, with appropriate and high quality methodology, analysis and conclusions. Field survey based veg analyses are hugely time-consuming and as a result not nearly enough of our globally exceptional flora is delineated using evidence based methods and this paper sets an excellent standard, and provides a solid foundation for delineation of better units, unit descriptions and mapping of boundaries than the original substrate+expert 2006 SA Vegmap units. The approach also aims to not disrupt existing boundaries where not justified. Should other experts disagree with any delineation, background, methods and results are transparently presented supporting consensus and resolution.
I have made a number of clarity/style suggestions below which I would consider optional and found a few minor typos. I would be comfortable with accepting without any required changes other than the following two issues which I think require author review/clarification:
1) Critical issue: 463 South Coast Strandveld concept and discussion in relation to Vegmap terminology
466-468 I’m very confused here – There does not seem to be such a thing as 466 Rebelo et al 2006 South Strandveld Bioregion. Rebelo 2006 mentions South Coast Strandveld, but only describes South Coast Strandveld briefly on page 76 (just before their section 3.4 Fynbos Thicket) indicting Moll et al 1984 dichotomy between West Coast Strandveld and South Coast Strandveld and that their classification “builds on this” which doesn’t seem to actually have been incorporated in the vegmap bioregions or concepts - the only bioregion within the Fynbos Biome section described with units allocated is for Western Strandveld (summary table Fynbos section page 55) – I can find no set of units assigned to a South Coast Strandveld bioregion here or elsewhere in Rebelo section or overall Mucina et al 2006 Vegmap. Other coastal units corresponding to a Strandveld or Dune veld concept are within other major biomes or bioregions as noted on line 468-9 for Algoa Dune Strandveld. This seems to be a poor naming convention within the original vegmap publication (western is indeed an odd choice for units extending onto the South Coast – and there is a nod to this on page 198 as the final para for their section 9.3 Western Strandveld.). Nonetheless, please confirm and clarify as this has an important implication – in that this publication is proposing defining and new and separate bioregion separate from Western Strandveld, and the key floristic differences at bioregion level then need to be addressed. My confusion is furthered by use of West Strandveld on line 488
2) Less important conceptual issue
As per conclusion 979 The vegetation of the Holocene dunes of the Cape south coast form a coherent geo-botanical unit, “
However no other non-Strandveld or more western Strandveld survey areas were included in analyses, so this conclusion of vegetation coherence should be presented in that context. That does not take anything away from the very high value of surveying, analysing and redefining the units within the study area, but may unduly influence future work aligning or separating surrounding areas. E.g. it is possible that Cape Peninsula and surrounding units when properly surveyed and compared to this data set, may align more closely with the presented South Coast Strandveld concept than the existing Western Strandveld concept.
Request – upload electronic GIS files of newly mapped units with publication resources.

---

## Round 0.2 · accepted · Accept

The authors have addressed the reviewers' comments satisfactorily. I have checked the response to the reviewers' comments and the track changes manuscript and this does not need to go past the reviewers again and am happy to accept this paper.